# CSN5i-3 is an orthosteric molecular glue inhibitor of COP9 signalosome

Huigang Shi[1], Xiaorong Wang[2], Clinton Yu[2], Haibin Mao[1,3], Fenglong Jiao[2], Merav Braitbard[4], Ben Shor[4], Zhongsheng Zhang[5], Thomas R. Hinds[1], Shiyun Cao[1,3], Erkang Fan[5], Dina Schneidman-Duhovny[4], Lan Huang[2✉] & Ning Zheng[1,3✉]

Orthosteric inhibitors block enzyme active sites and prevent substrates from binding[1]. Enhancing their specificity through substrate dependence seems inherently unlikely, as their mechanism hinges on direct competition rather than selective recognition. Here we show that a molecular glue mechanism unexpectedly imparts substrate-dependent potency to CSN5i-3, an orthosteric inhibitor of the COP9 signalosome (CSN). We first confirm that CSN5i-3 inhibits CSN, which catalyses NEDD8 (N8) deconjugation from the cullin-RING ubiquitin ligases, by occupying the active site of its catalytic subunit, CSN5, and directly competing with the iso-peptide bond substrate. Notably, the orthosteric inhibitor binds free CSN with only micromolar affinity, yet achieves nanomolar potency in blocking its deneddylase activity. Cryogenic electron microscopy structures of the enzyme–substrate–inhibitor complex reveal that active site-engaged CSN5i-3 occludes the substrate iso-peptide linkage while simultaneously extending an N8-binding exosite of CSN5, acting as a molecular glue to cement the N8–CSN5 interaction. The cooperativity of this trimolecular CSN5i-3–N8–CSN5 assembly, in turn, sequesters CSN5i-3 at its binding site, conferring high potency to the orthosteric inhibitor despite its low affinity for the free enzyme. Together, our findings highlight the modest affinity requirements of molecule glues for individual target proteins and establish orthosteric molecular glue inhibitors as a new class of substrate-dependent enzyme antagonists.

Enzymatic inhibitors are indispensable tools for dissecting biological pathways and developing therapeutic interventions[1]. They are broadly categorized by their binding sites and mechanisms of action. Among these, orthosteric inhibitors, which bind to the catalytic site and directly compete with substrates, have been extensively explored due to their predictable structure–activity relationships. However, such inhibitors are typically substrate-agnostic, as their mechanism relies solely on blocking the active site. By contrast, substrate-dependent inhibitors, which achieve selectivity by engaging allosteric sites or exosites, can modulate enzyme activity in a substrate-specific manner[2]. Yet their design remains challenging due to the complex structural and dynamic determinants governing these interactions. Ideally, combining the tractability of orthosteric inhibitors with the precision of substrate-dependent modulation would offer a powerful strategy—but whether such a hybrid approach is feasible has remained unclear.

The CSN is a multi-subunit protein complex evolutionarily conserved across all eukaryotic species[3,4]. It plays a crucial role in regulating almost every aspect of cellular functions by modulating the cullin-RING ubiquitin ligases (CRLs) and is required for targeted protein degradation induced by emerging protein degraders[5–12]. As the largest family of E3s, the function of CRLs hinges on the dynamic modification of the cullin scaffolds by a ubiquitin-like protein, NEDD8 (N8), also known as

neddylation[13]. Although cullin neddylation enhances the E3 activity of CRLs, CSN-mediated cullin deneddylation is thought to protect the CRL substrate receptors from auto-ubiquitination and promote their exchange on the cullin scaffolds through an adaptive CRL assembly cycle[14,15].

COP9 signalosome comprises eight core subunits (CSN1–CSN8), including the catalytic subunit CSN5 (Fig. 1a)[16]. The crystal structure of the CSN holoenzyme revealed that CSN5 and CSN6 form a heterodimer, which is affixed to the other six subunits[17]. The active site of CSN5 is occluded by its insertion-1 (Ins-1) loop in the CSN holoenzyme structure, suggesting that the iso-peptidase complex might be in an inactive state. Multiple subsequent cryogenic electron microscopy (cryo-EM) studies have captured the overall architecture of CSN in complexes with four different N8-CRLs[18–22]. These structures revealed a catalytic hemisphere of CSN formed among CSN2, CSN4, CSN5 and CSN6 that undergoes major conformational changes following substrate engagement (Fig. 1b). All of these structures, however, suffer from a limited resolution, especially within the catalytic hemisphere. How the N8-cullin iso-peptide linkage is recognized by the catalytic site of CSN in its pre-catalytic state remains to be elucidated.

CSN5i-3 has recently been developed as a small molecule CSN5 inhibitor that can potently block cullin deneddylation by CSN (Fig. 1c)[23].

[1]Department of Pharmacology, University of Washington, Seattle, WA, USA. [2]Department of Physiology and Biophysics, University of California, Irvine, CA, USA. [3]Howard Hughes Medical Institute, University of Washington, Seattle, WA, USA. [4]The Rachel and Selim Benin School of Computer Science and Engineering, The Hebrew University of Jerusalem, Jerusalem, Israel. [5]Department of Biochemistry, University of Washington, Seattle, WA, USA. ✉e-mail: lanhuang@uci.edu; nzheng@uw.edu

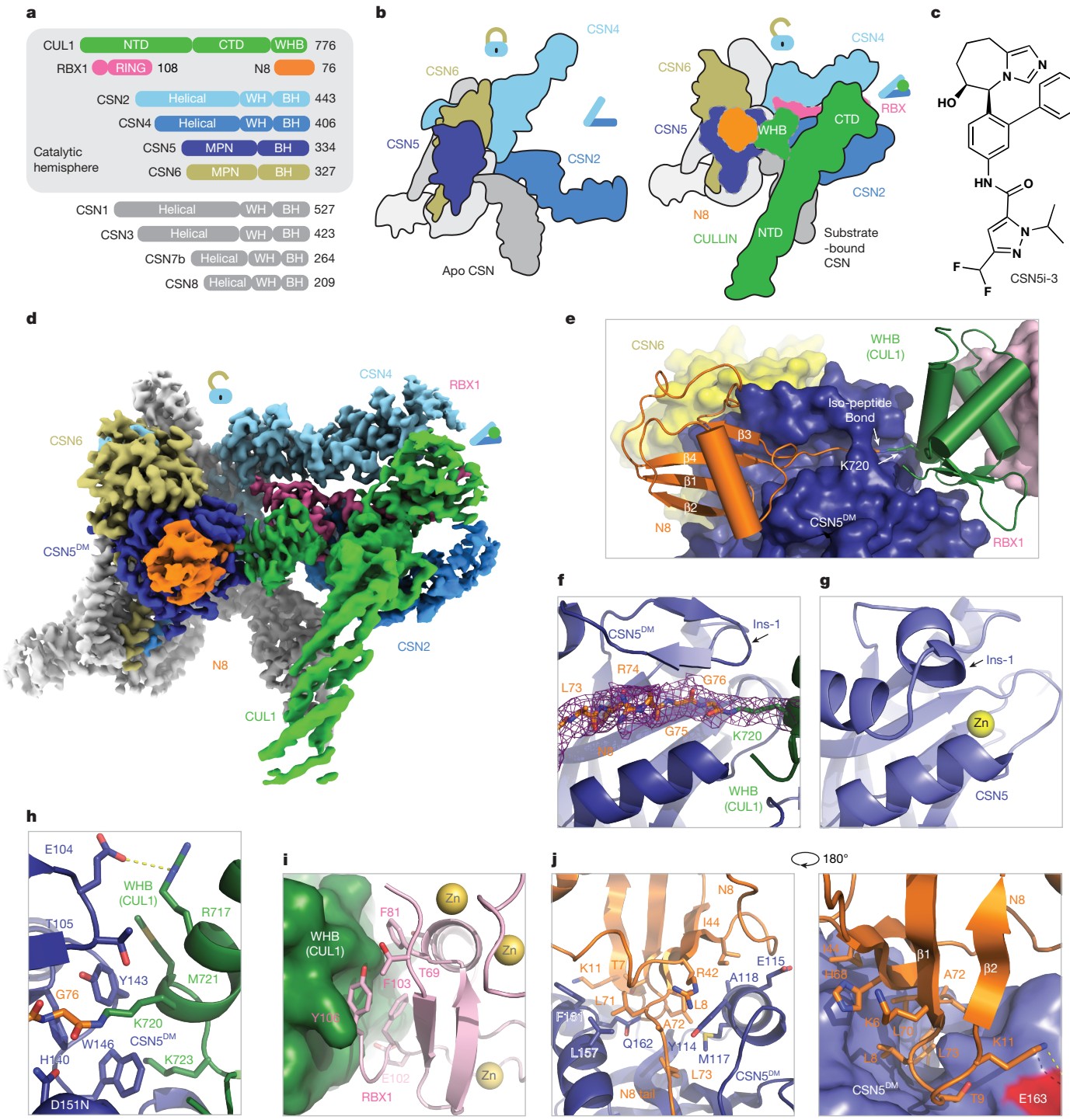

**Fig. 1 | The pre-catalytic state of CSN in complex with N8–CRL1. a**, Domain composition of CSN subunits, N8 and the CUL1–RBX1 E3 scaffold. **b**, Schematic drawings of CSN in its apo and N8-cullin-RING-bound forms with the CSN subunits in the catalytic hemisphere and the N8-CRL substrate coloured. Major conformational changes in CSN following substrate engagement are indicated by lock and clamp icons. **c**, The chemical structure of CSN5i-3. **d**, Cryo-EM map of the CSN$^{DM}$–N8-CRL1 complex. **e**, The three-molecular interface among CSN5$^{DM}$ (surface in dark blue), N8 (cartoon in orange) and the CUL1–WHB domain (carton in green). The iso-peptide bond formed between the N8 C-terminal carboxyl group and the lysine residue of CUL1 (Lys120) is indicated. **f**, Recognition of the iso-peptide bond formed between N8 and Lys720 of the CUL1–WHB domain by CSN5$^{DM}$ with its reshaped Ins-1 loop. **g**, A close-up view of the catalytic site in isolated CSN5 occupied by its Ins-1 loop. PDB ID: 4D10. **h**, A close-up view of the interface between CSN5$^{DM}$ and the CUL1–WHB domain with the substrate's iso-peptide bond housed at the catalytic site. **i**, A close-up view of the WHB–RBX1 interface. **j**, Two close-up views of the exosite interface between N8 (orange) and CSN5$^{DM}$ (slate).

Although the compound directly targets the CSN5 active site, it has recently been reported to inhibit the deneddylase complex via a surprisingly uncompetitive mechanism[24]. Unlike a non-competitive inhibitor, an uncompetitive (also known as anti-competitive) inhibitor is expected to bind only to the enzyme–substrate complex, not to the free enzyme[25]. Here we discover that CSN5i-3 unexpectedly acts as a molecular glue, which not only stabilizes the CSN–N8-CRL complex, but also gains its high potency in a substrate-dependent manner. This unusual

mechanism of action leads us to establish the concept of orthosteric molecular glue (OMG) inhibitors.

## Pre-catalytic state of CSN^DM–N8~CRL1

As a metalloprotease, CSN5 features a catalytic $Zn^{2+}$ ion at the end of a hydrophobic cleft, which has been predicted to recognize the C-terminal tail of N8 conjugated to cullins[18]. A crystal structure of isolated CSN5 bound to CSN5i-3 revealed that the compound directly coordinates the $Zn^{2+}$ ion and occupies the enzyme active site as an orthosteric inhibitor[23]. Before reconciling its orthosteric nature and its reported uncompetitive mechanism, we first aimed to experimentally determine how the N8-CRL1 iso-peptide linkage is physically engaged with the CSN5 catalytic cleft in the absence of the compound. The catalytic zinc ion in CSN5 is coordinated by an aspartate (Asp151) and two histidine residues (His138 and His140), which are joined by an upstream water-activating glutamate residue (Glu76)[16,17,26]. Using a catalytically impaired CSN mutant harbouring a CSN5 H138A mutation, past studies have not been able to resolve the binding mode of the N8-cullin conjugates at the CSN5 catalytic site. In an enzymatic assay with N8-CRL1 as a substrate, we found that the CSN^CSN5-H138A mutant, as well as several other documented CSN mutants, still retain a detectable iso-peptidase activity (Extended Data Fig. 1a). By contrast, a CSN5 E76A/D151N double mutation completely abolishes the catalytic activity of CSN towards N8-CRL1. Leveraging this CSN mutant (hereafter referred to as CSN^DM), we determined the cryo-EM structure of a CSN^DM–N8-CRL1 complex at 3.2 Å resolution (Fig. 1d, Extended Data Fig. 1b and Extended Data Table 1).

Distinct from all previously reported CSN–N8-CRL structures, the catalytic hemisphere of the CSN^DM–N8-CRL1 complex is clearly resolved in the three-dimensional reconstruction map with the substrate iso-peptide linkage stably trapped at the CSN5 catalytic site (Fig. 1e). The resulting structural model confirms the previously described global architectural changes in CSN following the binding of N8-CRLs; such changes are highlighted by the pivotal movement of the CSN5–CSN6 dimer, which is induced by the association of CUL1-CTD–RBX1 with CSN2 and CSN4. The complex structure also reveals a cascade of protein–protein interactions around the catalytic centre, involving N8, CSN5, CUL1–WHB and the RBX1 RING domain. Importantly, the resolution of the structure is high enough to resolve the complete N8 C-terminal tail conjugated to the side chain of the CUL1–WHB Lys720 residue, which together span the entire CSN5 catalytic cleft. In comparison to the apo form of CSN, the central region of the CSN5 Ins-1 loop is dislodged from the catalytic cleft and adopts a β-hairpin structure, forming a three-stranded anti-parallel β-sheet with N8 C-terminal tail (Fig. 1f,g). Such a structural arrangement positions the N8-CRL1 iso-peptide bond right above the catalytic zinc ion binding site, representing the pre-catalytic state of the enzyme–substrate complex.

## Pre-catalytic state protein interactions

Similar to all previously reported CSN–N8-CRL structures, the polypeptide sequence connecting the CUL1 WHB domain to the CTD has no visible density and is presumably flexible in structure. The CUL1 WHB domain, nevertheless, is well resolved in the cryo-EM map, forming an extensive interface with CSN5 centred around the neddylation site, CUL1-Lys720 (Fig. 1h and Extended Data Fig. 1c). Part of this interface is made between CUL1–WHB and the CSN5 MPN core domain, where the aliphatic side chain of CUL1-Lys720 is buttressed by two aromatic residues of CSN5, Tyr143 and Trp146. These two CSN5 residues, at the same time, interact with Met721 and Lys723 of CUL1–WHB through hydrophobic and π–cation interactions, respectively. Two further CSN5 residues at the tip of the Ins-1 loop β-hairpin, Glu104 and Thr105, further strengthen the interface by interacting with Arg717 of CUL1–WHB via

a salt bridge and van der Waal packing (Fig. 1h). Following engaging N8-CRL1, the rotation movement of the CSN5–CSN6 dimer pivots the CSN5 Ins-2 loop away from the helical bundle. Adopting a new conformation, this second CSN5 insertion loop joins Trp146 to wrap around the aliphatic side chain of CUL1-Lys723 (Extended Data Fig. 1c). Two of its hydrophobic residues—Ile211 and Val216—further augment the interface via hydrophobic packing.

To test the functional importance of this CSN5–CUL1–WHB interface, we measured the enzymatic activity of CSN with the CSN5 Tyr143 and Trp146 residues individually mutated to alanine. Both mutants were catalytically compromised, albeit to different degrees (Extended Data Fig. 1d). Similarly, an internal truncation of the CSN5 Ins-2 loop (Δ205–217), markedly attenuated the activity of the iso-peptidase complex. In fact, converting Val216, a hydrophobic residue in the Ins-2 loop, to a charged amino acid was sufficient to achieve the same effect. These results suggest that the CSN5–CUL1–WHB interaction is functionally coupled with the engagement of the iso-peptide linkage at the CSN5 catalytic cleft.

By securing the CUL1 WHB domain against CSN5, the active site-engaged iso-peptide linkage also stabilizes the packing of CUL1–WHB against the RBX1 RING domain at the opposite side (Fig. 1i). Characterized by a mixture of hydrophobic and polar interactions, this CUL1–WHB–RBX1-RING interface is mediated by a cluster of three aromatic residues—Phe81, Phe103 and Tyr106—which are strictly conserved in RBX1 orthologues (Extended Data Fig. 1e). Mutation of these residues to either serine or alanine effectively abolished the enzymatic activity of CSN without impacting N8-CRL1 binding (Extended Data Fig. 1f–i). The optimal enzymatic activity of CSN therefore requires a continuum of intermolecular interfaces extending from the CSN5 active site to the RBX1 RING domain, which itself is locked by CSN2 and CSN4 as shown by a past study[18].

## N8-binding exosite on CSN5

Aside from the two separate interfaces made by the CUL1–WHB domain with CSN5 and the RBX1 RING domain, immobilization of the iso-peptide conjugate at the CSN5 active site also stabilizes the N8 globular domain, which is anchored to an exosite of CSN5 next to the catalytic cleft.

Immediately preceding the six amino acids N8 C-terminal tail (Leu71–Gly76), the β1–β2 loop of the N8 globular domain is wedged into a surface pocket of CSN5 adjacent to its catalytic cleft (Fig. 1j). On the two sides of this loop are two overall hydrophobic patches (including one centred around Ile44) that mediate the recognition of N8 by CSN5. The tip of the loop is further locked in place by a network of hydrogen bonds and salt bridges. A missense mutation of a CSN5 glutamate residue at this interface (E163 in human CSN5) causes defects in photoreceptor neuron projections in *Drosophila*[27]. Despite anchoring at a different angle, the binding mode of the N8 globular domain to CSN5 resembles the recognition of the distal ubiquitin by AMSH-LP, a JAMM-type zinc-dependent deubiquitinase capable of cleaving Lys63-linked polyubiquitin chain (Extended Data Fig. 2a)[26]. All residues in the two modifiers that are perceived by their cognate iso-peptidases are conserved between N8 and ubiquitin, even though the majority of the modifier-binding amino acids are different in the two enzymes (Extended Data Fig. 2b). This exosite interface between N8 and CSN5 is therefore unlikely to be the main determinant of the substrate specificity of CSN. In agreement, CSN has been previously shown to be able to hydrolyse the artificial ubiquitin–rhodamine substrate[17]. Although this N8–CSN5 exosite interface cannot differentiate N8 from ubiquitin, it is crucial for cullin deneddylation. Mutations of the primary N8-contacting residues in CSN5 (that is, Tyr114, Phe161 and Gln162) impair or even abrogate the enzymatic activity of CSN (Extended Data Fig. 1d). The productive engagement between the deneddylase complex and its substrate therefore entails a network

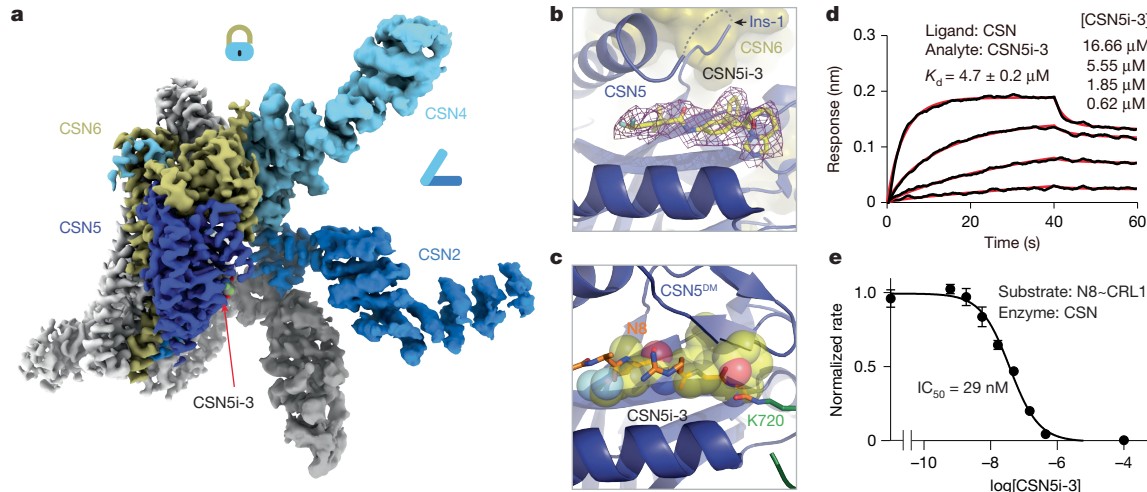

**Fig. 2 | CSN5i-3 as a hybrid orthosteric uncompetitive inhibitor of CSN.**
**a**, Cryo-EM map of CSN5i-3-bound CSN with subunits in the catalytic hemisphere coloured and the inhibitor shown in spheres. **b**, A close-up view of CSN5 catalytic cleft occupied by CSN5i-3, the density of which is shown in purple mesh. **c**, Steric hindrance imposed by CSN5i-3 for the engagement of the N8-CRL1 iso-peptide linkage at the CSN5 active site. **d**, The binding affinity of CSN5i-3 to CSN measured by BLI. **e**, The potency of CSN5i-3 in inhibiting the iso-peptidase activity of CSN towards N8-CRL1 in vitro. Data are mean ± s.d. of $n = 3$ biological replicates.

of highly coupled protein–protein interactions traversing from the N8-binding exosite on CSN5 to CSN2/4 via the iso-peptide linkage, the CUL1 WHB domain, and the RBX1 RING domain.

## Pre-catalytic states of all CSN–N8~CRLs

To confirm whether the pre-catalytic architecture of CSN–N8-CRL1 is applicable to other CRLs, we obtained the cryo-EM structures of CSN$^{DM}$ in complexes with N8-modified CUL2-RBX1, CUL3-RBX1, CUL4A-RBX1 and CUL5-RBX2. Apart from the CSN$^{DM}$–N8-CRL3 structure, which was resolved at a 3.9 Å resolution, all other structures were determined with a high enough resolution (3.0–3.5 Å) to reveal the detailed interfaces within the catalytic hemisphere (Extended Data Fig. 2c–f and Extended Data Table 1). As expected, these structures share the same topology as the CSN$^{DM}$–N8-CRL1 complex, which is highlighted by the same chain of interactions that bridges N8 and CSN2/4. The three aromatic residues of RBX1 interfacing with the CUL–1WHB domain are replaced by three non-aromatic hydrophobic residues in RBX2 (Extended Data Fig. 2g). The same RING-WHB binding pose is nevertheless retained in the CSN$^{DM}$–N8-CRL5 complex. A superposition analysis reveals that the WHB domains of CUL1 and CUL5 are positioned slightly differently relative to their flanking CSN5 and RING partners (Extended Data Fig. 2h). Such a topological variation is accommodated by the differential pivotal angles of the CSN5–CSN6 dimer related to the rest of CSN. The iso-peptidase complex therefore harnesses both its intrinsic plasticity and synchronized protein-protein interactions to catalyse the deneddylation reaction across all cullin proteins.

## CSN5i-3 as a unique orthosteric inhibitor

After revealing how the substrate and its iso-peptide interact with the CSN active site, we next determined the CSN5i-3–CSN complex structure by cryo-EM at 3.3 Å resolution to verify the binding mode of the inhibitor in the context of the intact deneddylase complex (Fig. 2a, Extended Data Fig. 3a and Extended Data Table 2). The structure of CSN bound to CSN5i-3 is nearly identical to its apo form, except that the inhibitory compound is sufficient to displace the Ins-1 loop and gain access to the active site without the engagement of a neddylated cullin-RING substrate (Fig. 2b). CSN5i-3 adopts the same binding pose on CSN5 within the CSN complex as it does on isolated CSN5. When superimposed with the CSN$^{DM}$–N8-CLR1 structure, CSN5i-3 would

severely clash with the N8 C-terminal tail, the iso-peptide bond and the CUL1–Lys720 side chain, confirming its orthosteric nature (Fig. 2c).

To further characterize the inhibitor, we used biolayer interferometry (BLI) and measured the affinity of CSN5i-3 to a purified CSN5–CSN6 heterodimer. Strikingly, the dissociation constant ($K_D$) of the inhibitor–CSN5–CSN6 interaction was determined to be around 4 μM (Extended Data Fig. 3b), which differs from the originally reported potency of CSN5i-3 (~5.8 nM) by three orders of magnitude[23]. This unexpected moderate affinity of the inhibitor was further verified by isothermal titration calorimetry (ITC) (Extended Data Fig. 3c). Consistent with its identical binding mode on isolated and CSN-embedded CSN5, the inhibitor displayed a similar affinity (~4.7 μM) towards the free eight-subunits CSN complex (Fig. 2d). Using N8-CRL1 as a substrate, we further confirmed that CSN5i-3 inhibited the CSN complex for deconjugating N8-CRL1 with a low nanomolar IC$_{50}$ (29 nM) (Fig. 2e). CSN5i-3 therefore acts as a non-canonical uncompetitive inhibitor that preferentially binds to the enzyme–substrate complex rather than to the free enzyme. Based on the established model of CSN activation, the most parsimonious explanation would be substrate-induced remodelling of the auto-inhibitory Ins-1 loop, which might facilitate CSN5i-3 binding to the CSN5 active site.

## Structure of CSN5i-3-bound CSN–N8~CRL1

To decipher the precise uncompetitive mechanism of CSN5i-3, we assembled and determined the cryo-EM structure of inhibitor-bound CSN in complex with N8-CRL1 at 3.0 Å resolution (Fig. 3a, Extended Data Fig. 3d and Extended Data Table 2). As expected, when bound to the N8-CRL1 complex, CSN5i-3 blocks the CSN5 catalytic cleft and pushes the substrate iso-peptide conjugate out of the active site. Although such an action of the orthosteric inhibitor is anticipated to disrupt the protein interaction network that stabilizes the pre-catalytic state of the enzyme–substrate assembly, the CSN–N8-CLR1–CSN5i-3 complex structure reveals a well-organized catalytic hemisphere with clearly resolved densities for N8, CUL1–WHB and RBX1-RING. Although the orthosteric inhibitor prevents the substrate iso-peptide linkage from binding the CSN5 active site, it seems to make direct contact with N8 and CUL1–WHB, stabilizing the overall topology of the pre-catalytic assembly (Fig. 3b–d).

In complex with CSN–N8-CRL1, the elongated CSN5i-3 compound adopts the same binding pose as observed in the free form of CSN,

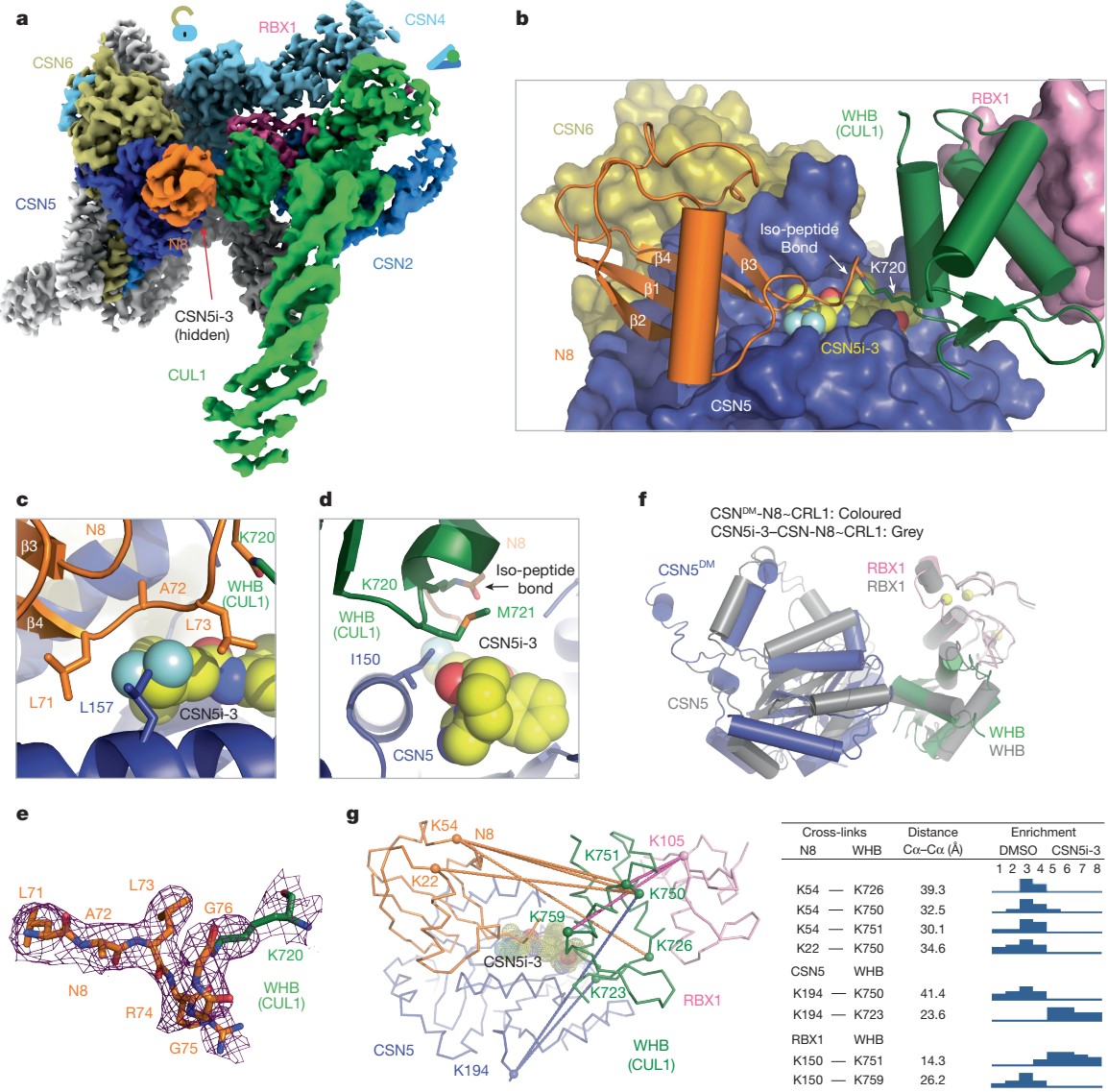

**Fig. 3 | Stabilization of CSN–N8~CRL1 pre-catalytic state by CSN5i-3.**
**a**, Cryo-EM map of the CSN5i-3–CSN–N8~CRL1 complex with the substrate and CSN subunits in the catalytic hemisphere coloured. **b**, The four-molecular interface among CSN5i-3 (sphere), CSN5 (surface in dark blue), N8 (orange) and the CUL1–WHB domain (green). The iso-peptide bond formed between the N8 C-terminal carboxyl group and the lysine residue of CUL1 is indicated. **c**, A close-up view of the interface between CSN5i-3 and the C-terminal tail of N8. **d**, A close-up view of the interface among CSN5i-3, CSN5 and the CUL1–WHB domain. **e**, The cryo-EM density of the iso-peptide bond flanked by the N8 C-terminal tail and the side chain of the neddylation site lysine residue in CUL1–WHB domain. **f**, Positional difference of CSN5 relative to the CUL1–WHB domain anchored to the RBX1 RING domain in the structures of CSN^DM and CSN5i-3-bound CSN in complex with N8~CRL1. **g**, Select cross-links among N8, CSN5, CUL1–WHB and RBX1-RING mapped on the CSN5i-3–CSN–N8~CRL1 cryo-EM structure along with the corresponding Cα–Cα distances and the enrichment profile detected by quantitative TMT mass spectrometry.

spanning the whole catalytic cleft and burying the catalytic zinc ion underneath (Extended Data Fig. 3e). At one end of the inhibitor, the difluoromethyl pyrazole moiety of CSN5i-3 packs closely against the N-terminal half of the N8 C-terminal tail, which consists of three residues: Leu71, Ala72 and Leu73 (Fig. 3c). The difluoromethyl group, in particular, is sandwiched between the two leucine residues of N8 and is further shielded by Leu157 of CSN5. Together, they nucleate a hydrophobic core and extend the adjacent N8–CSN5 exosite interface (Fig. 3b and Extended Data Fig. 3f).

At the opposite end of the catalytic cleft, the otherwise solvent exposed azepine group of the inhibitor is engaged with Met721 from the CUL1 WHB domain, which is next to the neddylation site, Lys720. In conjunction with the nearby Ile150 residue in CSN5, they weld a separate trimolecular hydrophobic interface (Fig. 3d). Although excluded from the catalytic cleft by CSN5i-3, the linkage between the C-terminal

diglycine motif of N8 and the side chain of CUL1 Lys720 is clearly visible in the cryo-EM map, overarching the inhibitory compound (Fig. 3b,e). Compared with the CSN^DM–N8~CRL1 structure, the CUL1 WHB domain maintains its interface with the RBX1 RING domain while packing against CSN5 at a slightly different angle (Fig. 3f).

To verify the topology of the catalytic hemisphere of the CSN–N8~CRL1 complex stabilized by CSN5i-3, we used the quantitation of multiplexed, isobaric-labelled cross-linked peptides (QMIX) strategy for multiplexed quantitative cross-linking mass spectrometry (XL-MS) analysis of CSN mixed with N8~CRL1 in the presence and absence of the inhibitor[28]. A mass-spectrometry-cleavable lysine-reactive cross-linker, disuccinimidyl sulfoxide (DSSO), was coupled with isobaric tandem mass tag (TMT) reagents for quantitative profiling of protein–protein interactions with residue-specific resolution[29]. As expected, CSN5i-3 treatment enriched numerous intermolecular

cross-links between CUL1 and N8, probably by blocking the deneddylation reaction. A few cross-links between the CUL1–WHB domain and N8, nevertheless, were clearly suppressed by the inhibitor, consistent with their spatial separation as observed in the cryo-EM structure (Fig. 3g and Supplementary Table 1). Remarkably, CSN5i-3 elicited opposite effects on the cross-links of a lysine residue in RBX1-RING as well as in CSN5 to two lysine residues in the CUL1–WHB domain. In both cases, the opposite effects can be explained by the maximal inter-lysine distance (~30 Å) allowed for efficient cross-links (Fig. 3g). Our XL-MS data therefore fully support the spatial architecture of the enzyme–substrate complex resolved by cryo-EM.

## CSN5i-3 as a molecular glue

Despite acting as an orthosteric inhibitor, CSN-bound CSN5i-3 retains the N8–CUL1–WHB conjugate, which seems to block its exit path and trap it in the catalytic cleft. The interfacial nature of the compound hints at a cooperative binding mechanism characteristic of a molecular glue[30]. Specifically, the close proximity between CSN5i-3 and the N8–CSN5 exosite interface strongly suggests that the iso-peptidase inhibitor might play a role in extending the interface between the globular domain of N8 and CSN5, thereby enhancing their basal interaction. In return, the strengthened binding between CSN5 and the N8-CRL1 conjugate might stabilize the inhibitor and prolongs its retention at the CSN5 active site. Such a mechanism provides an alternative explanation for its substrate-dependent potency of the orthosteric inhibitor.

To test this idea, we first assessed the potency of CSN5i-3 in inhibiting the intrinsic catalytic activity of the CSN5–CSN6 heterodimer towards N8–rhodamine. Remarkably, the CSN inhibitor blocked the reaction with an $IC_{50}$ of 145 nM (Extended Data Fig. 3g). Such a high potency is in stark contrast to its micromolar affinity of binding free CSN5–CSN6 and is close to its potency in inhibiting N8-CRL1 deneddylation by the intact CSN (Fig. 2e and Extended Data Fig. 3b,c). The unique discrepancy between the high potency and the moderate affinity of the compound towards CSN, therefore, can be recapitulated with the much-simplified enzyme–substrate system and is largely independent of CRL1. In support of this conclusion, CSN5i-3 can also inhibit the hydrolysis of N8–rhodamine by CSN with a 95 nM potency (Extended Data Fig. 3h).

We next investigated whether the inhibitor can indeed promote the enzyme–substrate interaction as a molecular glue. As CSN5i-3 physically occludes the CSN5 active site and does not directly contact the iso-peptide bond formed between N8 and CUL1 Lys-720, we replaced N8-rhodamine with native N8 and measured its affinity toward the CSN5–CSN6 heterodimer in the presence and absence of CSN5i-3. Although the interaction between N8 and CSN5–CSN6 was barely detectable by BLI, they acquired a high affinity of ~930 nM when CSN5i-3 was present at a saturating concentration (Fig. 4a). Importantly, the same observation was made with the intact CSN, which can capture free N8 with an ~130 nM affinity in an inhibitor-dependent manner (Fig. 4b). Together, these data allowed us to conclude that the orthosteric inhibitor can indeed act as a molecular glue to seize N8 conjugated to CUL1 while excluding the iso-peptide conjugate from the catalytic cleft.

To further validate the conclusion, we determined the cryo-EM structure of CSN5i-3-bound CSN in complex with N8 at 3.3 Å resolution (Extended Data Fig. 4a and Extended Data Table 2). As expected, without CRL1 loaded, CSN in the complex adopts the same overall topology as revealed for its apo and CSN5i-3-bound forms (Fig. 4c). The presence of the orthosteric inhibitor, however, stabilizes the interaction between CSN and N8, which becomes well resolved in the cryo-EM map with clear side-chain densities (Extended Data Fig. 4a). A superposition analysis of the CSN5i-3–CSN–N8 and CSN5i-3–CSN–N8-CRL1 structures unveils a nearly identical binding mode of N8 on CSN5, albeit the differential orientations of the iso-peptidase subunit relative to the rest of the complex (Fig. 4d and Extended Data Fig. 4b). Although the C-terminal tail of N8 is excluded from the CSN5 catalytic cleft, it is shaped into a specific

coiled conformation by CSN5i-3 (Fig. 4e). Remarkably, removal or single amino acid mutations of the N8 C-terminal tail effectively abrogated the enhanced N8–CSN interaction fostered by the inhibitor (Fig. 4f).

CSN5i-1a is a precursor of CSN5i-3 and lacks the difluoromethyl pyrazole moiety that can directly interact with the N8 C-terminal tail (Fig. 4g). Analogous to the N8 C-terminal tail mutants, CSN5i-1a fails to promote the binding of N8 to CSN (Fig. 4h). Moreover, it inhibits CSN only with a single-digit micromolar potency, even though it docks to the CSN catalytic module with an affinity and binding pose similar to CSN5i-3 (Fig. 4i,j and Extended Data Fig. 4c). The distinction between the two compounds is further manifested by their differential effects on the binding affinity of N8-CRL1 to the deneddylase complex they orthosterically inhibit. In the presence of a saturating amount of compound, CSN interacts with its substrate with a $K_D$ of 1.8 µM and 26 nM, when being blocked by CSN5i-1a and CSN5i-3, respectively (Extended Data Fig. 4d,e). Together, these results reinforce the notion that CSN5i-3 strengthens the otherwise weak interaction between N8 and CSN5 as a molecular glue and attains its high potency by leveraging the cooperativity of the inter-molecular interactions at the trimolecular junction.

## Alteration of CSN interactome by CSN5i-3

As an uncompetitive inhibitor, CSN5i-3 shows a nearly irreversible effect in the cell, analogous to a covalent inhibitor[24]. Such a property suggests that the molecular glue effect of the compound might alter the dynamic of CSN–CRL interactions in the cell, which in turn affects its dissociation from the target. To assess the impact of CSN5i-3 on the interactome landscape of CSN, we used in vivo cross-linking-assisted affinity purification mass spectrometry to map the endogenous CSN-centric protein–protein interactions in the native cellular environment[31]. Two HEK293 stable cell lines that individually express HBTH-tagged CSN2 and CSN6 were used to isolate endogenous CSN complex and its interacting partners[32]. To capture both stable and dynamic CSN-interacting proteins, we performed in vivo chemical cross-linking before cell lysis. The purified protein complexes were digested and analysed by liquid chromatography tandem mass spectrometry (LC MS/MS) with both data-dependent (DDA) and data-independent (DIA) acquisitions. These analyses allowed us to identify not only all cullin proteins, RBX1/2 and all cullin adaptors (SKP1, ELONGIN-B/ELONGIN-C and DDB1), but also 159 CRL substrate receptors that are associated with CSN in the cell (Fig. 5a and Supplementary Table 2a,b).

We next performed DDA- and DIA-based label-free quantitative analyses to compare the levels of CSN subunits and their interacting proteins in the purified complexes from CSN5i-3-treated and untreated cells (Supplementary Table 2a,b). We also performed parallel reaction monitoring (PRM)-based targeted quantitation to examine protein expression levels of CSN subunits, CRLs and 101 substrate receptors in cells before and after CSN5i-3 treatment (Supplementary Table 2c). As expected, the amount of CSN subunits remained largely constant, indicating that the integrity of the iso-peptidase complex was not perturbed by the compound (Fig. 5b and Supplementary Table 2d). Although CSN5i-3 treatment induced minimal changes in the overall abundance of all cullins and N8 in cells, it increased their association with CSN (Fig. 5b and Supplementary Table 2d). This could be attributable to the accumulation of N8-CRLs in the cell and possibly the inhibitor-induced stabilization of the enzyme–substrate assembly. Strikingly, the amount of different CRL substrate receptors co-purified with CSN exhibited a broad range of CSN5i-3-induced changes, varying from near-complete depletion to pronounced enrichment. At one end of the spectrum, a fraction of CRL substrate receptors, such as DDB2 and FBXO22, showed diminished interactions with CSN following CSN5i-3 treatment (Fig. 5b). Their total protein level, however, dropped prominently in the cell. These observations were confirmed by immunoblotting analysis (Extended Data Fig. 4f–h).

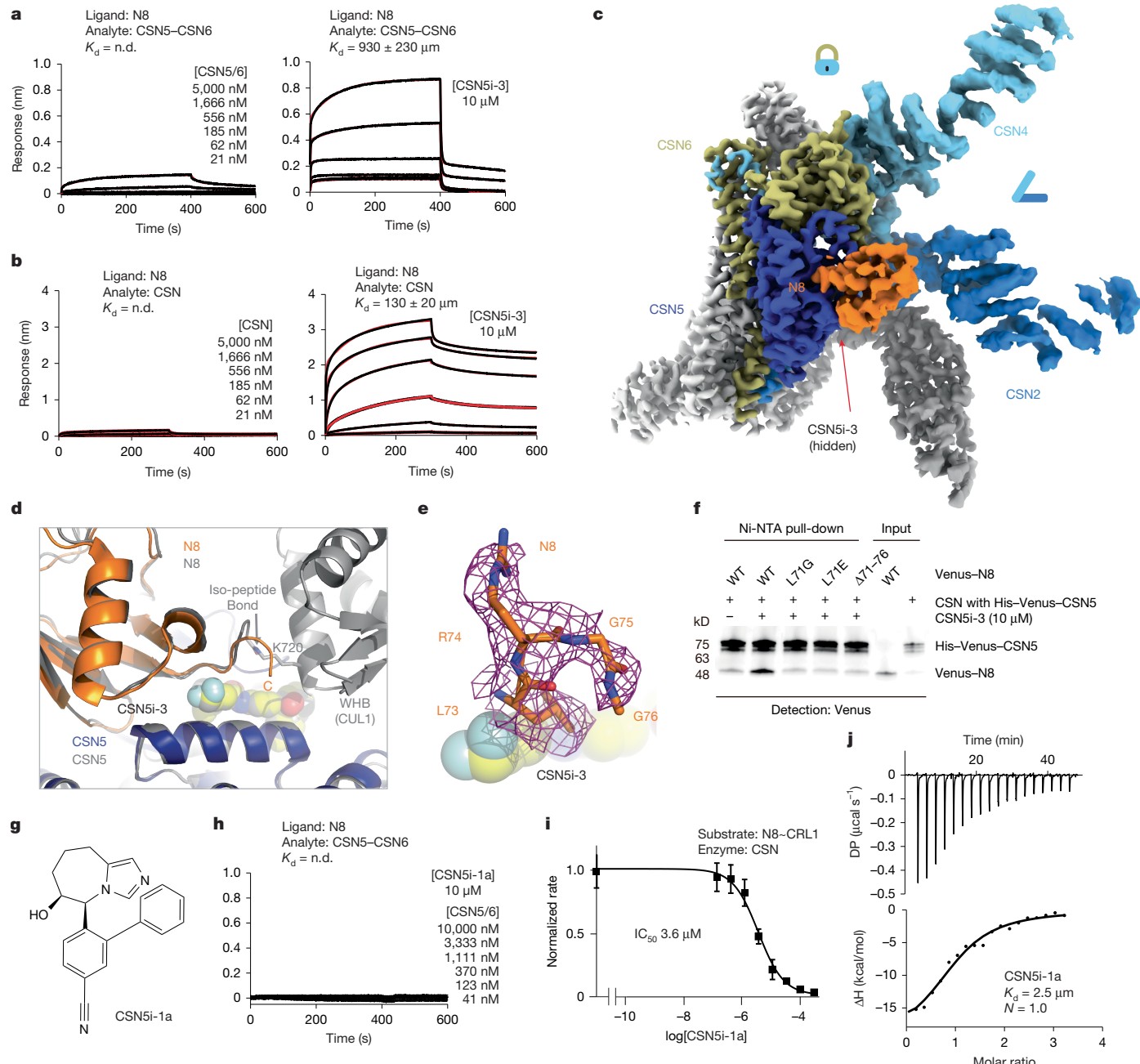

**Fig. 4 | CSN5i-3 acts as a molecular glue. a,b**, N8 binding to CSN5–CSN6 or CSN in the absence and presence of CSN5i-3 determined by BLI. **c**, Cryo-EM map of the CSN5i-3–CSN–N8 complex. **d**, A comparison of the N8 binding modes to CSN5 (left) in the CSN5i-3–CSN–N8–CRL1 and CSN5i-3–CSN–N8 complexes with CSN5 superimposed. **e**, Well-resolved density of the N8 C-terminal tail stabilized by CSN5i-3 in the CSN5i-3–CSN–N8 complex. **f**, The impact of N8 C-terminal tail truncation and mutations on CSN5i-3-enhanced N8–CSN interactions. The protein samples were not boiled before SDS–PAGE analysis (refer to Supplementary Fig. 1 for gel source data). **g**, The chemical structure of CSN5i-1a. **h**, CSN5i-1a shows no activity in promoting N8 binding to CSN. **i**, The potency of CSN5i-1a in inhibiting the iso-peptidase activity of CSN towards N8-CRL1 in vitro. Data are mean ± s.d. of $n = 3$ biological replicates. **j**, Binding affinity of CSN5i-1a to the CSN5–CSN6 heterodimer determined by ITC.

The overall destabilization of these CRL substrate receptors therefore corroborates with the proposed role of CSN in protecting them from auto-ubiquitination and subsequent degradation. At the other side of the spectrum, it is surprising that the majority of CRL substrate receptors such as DCAF4 and FBXL12 showed elevated association with CSN following inhibitor treatment even though their expression levels remained largely unchanged or was decreased (Fig. 5b and Extended Data Fig. 4f). This phenomenon—although contradictory to the hypothesized function of CSN—can be explained by the molecular glue activity of CSN5i-3 in locking the enzyme–substrate complexes. Past structural analyses have revealed that CSN subunits outside the catalytic hemisphere can make direct contacts with select CRL substrate receptors[4]. It is plausible that these interfaces can synergize with CSN5i-3 to differentially trap N8-CRLs with substrate receptors on CSN.

## Discussion

Here we discovered that CSN5i-3, a potent CSN inhibitor, functions dually as an orthosteric inhibitor of the iso-peptidase target while simultaneously stabilizing the enzyme–substrate complex as a molecular glue. Such an unexpected mechanism explains how the compound

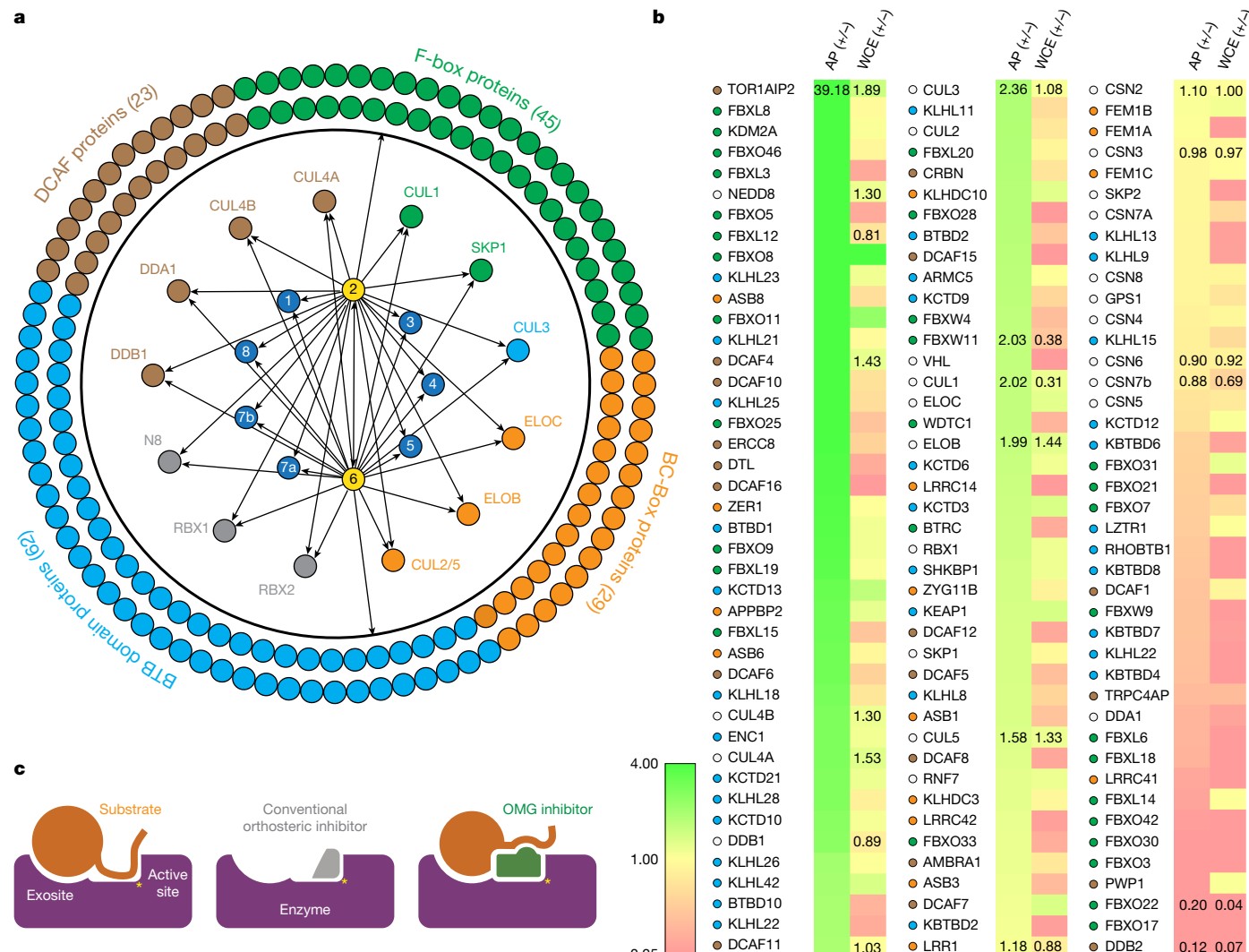

**Fig. 5 | CSN5i-3-induced changes in the CSN interactome. a**, CSN interaction network assessed by mass spectrometry analyses of affinity-purified (AP) CSN2 and CSN6 complexes. The central numbered circles represent the eight subunits of CSN. The CRL substrate receptor profiles for CSN2 and CSN6 affinity purification are indicated by arrows. **b**, CSN5i-3-induced abundance changes of selected proteins in affinity-purified CSN complexes and whole-cell extracts (WCE) determined by quantitative mass spectrometry analyses. Plus symbols, CSN5i-3-treated; minus symbols, untreated. **c**, Schematic drawing of the mechanism of action of an OMG inhibitor versus conventional orthosteric inhibitor. Left: the catalytic activity of an enzyme depends on substrate binding via the active site. Some enzymes also engage substrates through a

weakly interacting exosite that contributes to substrate recognition. Middle: conventional orthosteric inhibitors are designed to target the catalytic site of an enzyme and competitively occlude the substrate(s). These inhibitors are therefore substrate-agnostic. Right: an OMG inhibitor can be designed to achieve substrate-dependent potency by targeting the enzyme's active site with moderate affinity while making direct contact with a specific substrate. As a molecular glue, the OMG inhibitor stabilizes the interaction between the enzyme and a specific substrate. In return, the substrate strengthens the binding of the OMG inhibitor, thereby, affording the compound a potency that exceeds its affinity for the free enzyme.

inhibits N8-CRL deneddylation by CSN uncompetitively and gains its high potency in a substrate-dependent manner. We propose a new concept of orthosteric molecular glue (OMG) inhibitors (Fig. 5c) on the basis of this finding.

As an OMG inhibitor, CSN5i-3 possesses two unique properties that distinguish it from conventional orthosteric enzyme inhibitors. First, CSN5i-3 binds to the active site of its target with a moderate micromolar affinity but can inhibit the iso-peptidase complex with a low nanomolar high potency. This special attribute of OMG inhibitors points to an immediate implication in drug discovery. For instance, tailoring a canonical high-affinity orthosteric enzyme inhibitor to an OMG inhibitor with a substrate-dependent high potency could potentially overcome its off-target toxicity. Second, while blocking substrates from accessing the active site of an enzyme, an OMG inhibitor makes direct contact with a substrate protein to stabilize the enzyme–substrate

assembly. This unexpected property enables OMG inhibitors to achieve substrate selectivity, provided there are sequence variations at the substrate–OMG inhibitor interface. This feature is particularly valuable for enzymes whose disease-relevant versus toxicity-related substrates are expressed in different tissues, as it could allow selective targeting of pathological substrates while sparing those responsible for adverse effects.

Although the discovery of CSN5i-3 as an OMG inhibitor was fortuitous, we envision that CSN5i-3-like OMG inhibitors can be developed by design for many clinically relevant enzyme targets that recognize substrates proteins via exosites adjacent to the catalytic pockets. Such targets include intracellular and extracellular proteases, as well as enzymes catalysing the addition and removal of post-translational modifications. Rational development of OMG inhibitors can be guided by monitoring how derivative compounds of a canonical orthosteric

inhibitor affect enzyme–substrate interactions while enhancing catalytic potency.

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

## Methods

### Protein purification and complex assembly

Seven of the eight human CSN subunits, excluding CSN5, were overexpressed and purified from *Escherichia coli* (*E. coli*). Two trimeric subcomplexes, CSN1–2–3 and CSN4–6–7, were individually purified through co-expression. For the CSN1–2–3 subcomplex, CSN2 was subcloned into a modified pGEX4T1 vector (Amersham Biosciences) containing an N-terminal glutathione S-transferase (GST) tag followed by a tobacco etch virus (TEV) protease cleavage site. Truncated CSN3 (residues 1–409) and full-length CSN1 were individually subcloned into a modified pET15b vector (Novagen) containing an N-terminal His tag with a TEV protease cleavage site. The expression cassettes of CSN1 and CSN3$^{1-409}$—each driven by a T7 promoter and terminated by a T7 terminator—were subsequently inserted into the pGEX4T1-CSN2 plasmid. The resulting construct contained three independent expression cassettes for CSN2, CSN1 and CSN3$^{1-409}$, respectively. The CSN1–2–3 complex was co-expressed in *E. coli* BL21(DE3) cells (Novagen), purified by glutathione-affinity chromatography and subjected to TEV protease cleavage to remove the GST tag. The cleaved complex was further purified by anion-exchange (Source Q, Cytiva) and size-exclusion chromatography (Superdex 200, Cytiva). The CSN4–6–7 subcomplex was prepared using the same strategy. In this case, truncated CSN7b (residues 1–239) was subcloned into the modified pGEX4T1 vector, whereas full-length CSN4 and CSN6 were inserted into the modified pET15b vector. CSN8 was subcloned into a modified pET15b vector encoding an N-terminal maltose-binding protein (MBP) tag followed by a TEV protease cleavage site. The overexpressed MBP–CSN8 fusion protein was initially purified by MBP affinity chromatography, followed by on-column TEV protease cleavage to remove the MBP tag. The cleaved protein was then further purified by ion-exchange chromatography. The CSN-7mer complex—which comprises all subunits except CSN5—was reconstituted by incubating the CSN1–2–3, CSN4–6–7b and CSN8 a molar ratio of 1:1.5:2. The assembled complex was then purified by ion-exchange and size-exclusion chromatography.

The CSN5 subunit was prepared from insect cells using a Bac-to-Bac expression system. Wild-type and mutant CSN5 were subcloned into a modified GTE vector (Invitrogen) containing an N-terminal octa-histidine (8×His) tag, followed by a Venus fluorescent protein and a TEV protease cleavage site. High-titre baculovirus was generated using ExpiSF9 insect cells (Thermo Fisher Scientific) cultured at 27 °C. For large-scale protein production, ExpiSF9 cells at a density of $2 \times 10^6$ cells per millilitre were infected with baculovirus and cultured at 27 °C for 68 h. Cells were harvested by centrifugation, resuspended in lysis buffer containing 20 mM Tris-HCl pH 8.0, 200 mM NaCl, 1 mM PMSF and Roche Complete Protease Inhibitor and lysed by sonication. The lysate was clarified by centrifugation and CSN5 was purified through a series of chromatography steps, including His-tag affinity, ion-exchange and size-exclusion chromatography. To obtain untagged CSN5, the Venus tag was removed by TEV protease digestion performed during dialysis following the initial His-affinity purification. The resulting mixture was then passed through a second Ni-NTA column to separate untagged CSN5 from the cleaved His–Venus tag. To assemble the complete CSN complex, purified CSN5 was incubated with tag-free CSN-7mer at a 2:1 molar ratio on ice for 30 min and then purified by size-exclusion chromatography (Superdex 200) using a buffer containing 20 mM HEPES (pH 7.5), 150 mM NaCl and 0.2 mM TCEP. The peak fractions containing CSN complex were pooled, concentrated, flash-frozen and stored at −80 °C.

### Preparation of N8–CRL1 protein complexes

Two short unstructured segments in the N-terminus of CUL1 (residues 1–12 and 58–81) were removed from the full-length human CUL1, resulting in CUL1$^{\Delta N}$ (referred to here as CUL1). Both CUL1 and RBX1 (residues 16–108) were fused with an N-terminal 6xHis tag followed by a TEV protease cleavage site and co-expressed in BL21(DE3). The complex was initially purified using Ni$^{2+}$-Sepharose affinity chromatography. Following TEV cleavage to remove the His tag, the complex was further purified by cation-exchange and size-exclusion chromatography. The same strategy was used to prepare other CRL complexes, including CUL2-RBX1, CUL3-RBX1, CUL4A-RBX1 and CUL5-RBX2.

To prepare the heterodimeric N8-activating enzyme APPBP1–UBA3, APPBP1 was subcloned into the modified pGEX4T1 vector containing a GST tag followed by a TEV protease cleavage site, whereas UBA3 was subcloned into a modified pET15b vector containing a chloramphenicol resistance cassette. GST–APPBP1 and UBA3 were co-expressed in *E. coli* BL21(DE3) and purified by glutathione-affinity chromatography. After TEV cleavage, the APPBP1–UBA3 complex was further purified by anion-exchange and size-exclusion chromatography. The N8-conjugating enzyme UBC12 and N8 (wild-type and mutant forms) were subcloned into the same modified pGEX4T1 vector, expressed in *E. coli* BL21(DE3), and purified via glutathione-affinity and anion-exchange chromatography. A truncated form of N8 ending at glycine 76, representing the mature processed form, was used throughout this study. To generate the neddylated CRL1 complex (N8-CRL1), 10 µM of purified CRL1 was incubated with 10 µM GST–N8 in the presence of 0.2 µM APPBP1–UBA3 and 0.5 µM UBC12 at 4 °C for 1 h. The neddylation reaction was conducted in buffer containing 20 mM Tris-HCl pH 8.0, 200 mM NaCl, 0.4 mM TCEP, 2 mM ATP and 10 mM MgCl$_2$. The resulting GST–N8-CRL1 was separated from unmodified CRL1 by glutathione-affinity chromatography. After on-column TEV cleavage, N8-CRL1 was eluted from the column and further purified by cation-exchange and gel-filtration chromatography.

To conveniently monitor the production of free N8 during CSN-mediated deneddylation, N8 was site-specifically labelled with the fluorescent dye Alexa Fluor 633 (Sigma) via a cysteine residue engineered in place of the native N-terminal methionine. Excess dye was removed by size-exclusion chromatography. The fluorescently labelled N8 was then conjugated to the CUL–RBX1 complex using the neddylation protocol described above. For BLI experiments measuring the binding of N8 to CSN, an AviTag followed by a biologically inert GB1 tag (the B1 domain of streptococcal protein G; 56 residues, ~7 kDa) was fused to the N-terminus of N8. The purified Avi–GB1–N8 fusion protein was efficiently biotinylated in vitro using *E. coli* biotin ligase (BirA). Excess free biotin was removed by size-exclusion chromatography on a Superdex 75 column, yielding monodisperse, biotinylated Avi–GB1–N8 suitable for immobilization on streptavidin biosensors.

### Biolayer interferometry

The binding affinity between N8 and CSN or CSN5–CSN6 was measured using the Octet Red 96 system (Sartorius). Streptavidin (SA) biosensors (Sartorius) coated with streptavidin were loaded with 200 nM biotinylated N8 and then quenched with 200 nM biocytin before performing binding analyses. The reactions were conducted in black 96-well plates maintained at 30 °C. The binding buffer contained 20 mM HEPES pH 7.5, 150 mM NaCl, 0.2 mM TCEP, 0.1% Tween-20 and 0.1 mg ml$^{-1}$ ovalbumin. CSN or CSN5–CSN6 were tested as analytes at threefold serially diluted concentrations in the absence or presence of 20 µM CSN5i-3 (MedChemExpress). To assess the binding affinity between CSN$^{DM}$ and N8-CRL1 containing wild-type or 3A and 3S mutant RBX1, biotinylated N8-CRL1 was initially loaded to streptavidin-coated biosensors. CSN$^{DM}$ was tested as analyte at threefold serially diluted concentrations. To measure the binding affinity of CSN5i-3 with CSN or CSN5–CSN6, biotinylated anti-Venus nanobody was initially bound to streptavidin-coated biosensors to convert the probes to Venus-nanobody biosensors. Venus-tagged CSN was then loaded through the interaction between Venus and the anti-Venus nanobody. To measure the binding of CSN5i-3 to CSN5–CSN6, biotinylated CSN5–CSN6 protein was loaded to streptavidin-coated biosensors. CSN5i-3 were measured as analyte at threefold serially diluted concentrations. Data analysis of all BLI

experiments was performed using Octet data analysis software, and $K_D$ was determined from either kinetics or steady-state equilibrium measurements. All BLI experiments were performed a minimum of three times.

## Isothermal titration calorimetry

Isothermal titration calorimetry (Marven) binding assays were performed at 25 °C using a MicroCal PEAQ-ITC system (Malvern). To measure the binding of CSN5i-3 or CSN5i-1a to CSN5–CSN6, 160 μM CSN5–CSN6 in the syringe was titrated into 10 μM CSN5i-3 or CSN5i-1a in the cell. The data were fitted to a one-site binding model to determine the $K_D$, enthalpy ($\Delta H$) and stoichiometry ($N$) using the MicroCal PEAQ-ITC Analysis Software.

## Cryo-EM sample preparation and data collection

For grid preparation of all CSN complexes, holey gold grids (UltraAuFoil R1.2/1.3, 300 mesh) were pretreated with glow discharge. Fluorinated octyl maltoside (FOM) was added to all samples at a final concentration of 0.07% (w/v) before grid preparation. For CSN5i-3–CSN, CSN5i-3–CSN-N8, CSN5i-3–CSN–N8-CRL1, and CSN5i-1a-CSN complexes, 10 μM CSN was incubated first with 50 μM CSN5i-3 or CSN5i-1a and then with 30 μM N8 or N8-CRL1. Three microlitres of the sample were applied to grid. For the CSN$^{DM}$–N8-CRL (CRL1, CRL2, CRL3, CRL4A and CRL5) complexes, N8-CRLs were mixed with CSN$^{DM}$ at a 3:1 ratio and incubated for 30 min on ice. The concentration was diluted to approximately 4.5 mg ml$^{-1}$ for grid preparation. The grids were subsequently blotted for 6 s (blotting force = 0, temperature = 10 °C and relative humidity = 100%), plunged and flash frozen into liquid ethane using a Vitrobot Mark IV, and stored in liquid nitrogen for data collection.

Multiple complexes were imaged at different transmission electron microscopes at different Cryo-EM facilities. For the CSN, CSN$^{DM}$–N8-CRL4, CSN5i-3–CSN-N8 and CSN5i-1a–CSN complexes, data collection was performed on a Titan Krios transmission electron microscope (Thermo Fisher Scientific) operated at 300 kV at the University of Washington. For the CSN$^{DM}$–N8-CRL1, CSN$^{DM}$–N8-CRL2 and CSN5i-3–CSN–N8-CRL1 complexes, data collection was performed on a Titan Krios transmission electron microscope operated at 300 kV at the HHMI Janelia Research Campus. For CSN$^{DM}$–N8-CRL5 complex, data collection was performed on a Titan Krios transmission electron microscope operated at 300 kV at the National Cancer Institute. The automation scheme was implemented using the SerialEM software[33,34] with beam-image shift strategy and active beam-tilt compensation at a nominal magnification of 105,000×, resulting in a physical pixel size of 0.840 Å, 0.827 Å, 0.835 Å, respectively. The zero-loss-energy images were acquired on a Gatan K3 direct detector operated in correlated double sampling mode with the slit width of post-column Gatan Bio-Quantum GIF energy filter set to be 20, 20, 10-eV, respectively. The dose rate was adjusted to 13.9, 10.1, 14.6 e$^-$ per Å$^2$ per second, respectively, and a total dose of 60 e$^-$ per Å$^2$ for each image fractionated into 60 frames. The images were recorded at a defocus range of −0.8 - 3.0 μm. For the CSN$^{DM}$–N8-CRL3 and CSN5i-3–CSN complexes, data collection was carried out on a 200 kV Talos Glacios transmission electron microscope at University of Washington. The data collection performed with SerialEM automation scheme via beam-image shift strategy at a nominal magnification of 45,000x, resulting in a physical pixel size of 0.885 Å, using Gatan K3 direct detector operated in correlated double sampling mode. The dose rate was adjusted to 17.3 e$^-$ per Å$^2$ per second, and a total dose of 50 e$^-$ per Å$^2$ for each image fractionated into 100 frames. The images were recorded at a defocus range of −0.8–2.5 μm.

## Cryo-EM data processing

All datasets were processed with cryoSPARC[35]. Beam-induced motion correction and contrast transfer function (CTF) estimation were conducted using cryoSPARC Live session with Patch Motion Correction and patch CTF estimation. The motion-corrected micrographs were filtered by defocus, CTF fit resolution (excluding >8 Å) and relative ice thickness. The curated micrographs were exported and subject to blob picking. After inspection of picked particles and removal of targets on ice or Au foil, the left particles were extracted with four-fold binning and subject to two-dimensional classification. The good particles were kept and subject to ab initio reconstruction with four classes. All particles were then subject to heterogeneous refinement with the four reconstructions generated from ab initio reconstruction as reference. Approximately 100,000 particles were randomly selected after heterogenous refinement. Particles of good reconstructions were used as template to train model for Topaz picking[36]. After picked by topaz, particles were extracted and cleaned with two-dimensional classification. Good particles were kept and filtered by centre-to-centre distance to remove duplicate particles.

For CSN5i-3–CSN–N8-CRL1, 229,370 good particles from 7,564 micrographs generated from sequential blob picking, two-dimensional classification were subjected to sequential ab initio reconstruction and heterogenous refinement. We used 149,404 particles from good reconstruction for topaz training; 1,174,702 particles were picked using topaz picking. After one round of two-dimensional classification and duplicates curation, 536,325 particles were kept and subjected to heterogeneous refinement using the four reconstructions generated from previous ab initio reconstruction as reference. We kept 370,244 particles and subjected them to sequential non-uniform refinement and CTF refinement. These particles yielded a 2.9 Å reconstruction. Three-dimensional classification was applied to further improve the density of CSN5–CSN6. We kept 174,577 particles and yielded a 3.0 Å reconstruction. The following local refinement with a binary mask focused on CSN5–CSN6–N8-CUL1–WHB domain yields a 3.3 Å reconstruction.

For CSN$^{DM}$–N8-CRL1, 726,752 good particles from 10,052 micrographs were subjected to heterogenous refinement. We kept 434,896 particles from one good class at 5.2 Å for sequential particle extraction and non-uniform refinement[37]. After refinement, 433,043 particles yielded a nominal 3.3 Å reconstruction. The densities of CSN5–CSN6 and N8-CRL1 were less resolved than the C-terminal helical bundle of CSN. To further improve the density of the catalytic hemisphere, the particles were subject to three-dimensional variability analysis with a focused mask containing CSN5–CSN6 globular domain, N8, WHB and RBX1 globular domains implemented. After three-dimensional variability analysis[38], we kept 177,066 particles of improved density covering CSN5–CSN6 and N8-CUL1–WHB, which yielded a 3.2 Å reconstruction. Sequential global CTF, local and non-uniform refinement yielded a 3.2 Å reconstruction. The CSN5–CSN6–N8 density was further polished using the same binary mask as the three-dimensional variability imposed for local refinement. The local refinement improved the CSN5–CSN6 density and allowed us to visualize the side-chain density of N8, CSN5–CSN6 and the CUL1WHB domain. Refer to Extended Data Figs. 1, 3 and 4, as well as Extended Data Tables 1 and 2, for further information.

## Model building and refinement

For the CSN5i-3–CSN structure, the PDB model 4D10 was fitted into the 3.3 Å CSN-CSN5i-3 electron microscopy map using Chimera[39]. The fitted model was used as the initial structure to rebuild the CSN complex. The CSN7a from 4D10 served as a template to trace the main chain, and the residue sequence was modified to match CSN7b in Coot[40]. For the flexible regions of CSN2 and CSN4, the PDB model was trimmed based on the density in Coot. The CSN5i-3 model from 5JOG was fitted into the 3.3 Å CSN5i-3–CSN EM map and refined using Phenix. For the CSN5i-1a-CSN structure, CSN5i-1b in PDB model 5JOH was modified on the basis of a CSN5i-1a SMILES file. The CSN model from CSN5i-3–CSN and CSN5i-1a was then fitted into the 3.0 Å CSN5i-1a-CSN electron microscopy map and refined with Phenix. For the CSN5i-3–CSN–N8-CRL1 model, the previous CSN-CSN5i-3 structure was used as the starting model. The CSN5–CSN6, CSN2 and CSN4 subunits undergo conformational

changes upon binding to N8-CRL1. Their structural models were first split and then fitted into the 3.3 Å electron microscopy map. Residues were adjusted in Coot on the basis of sequence and side-chain density. The PDB model 1LDJ of CRL1 was fitted and manually adjusted in Coot. The iso-peptide bond was manually built in UCSF ChimeraX. For the CSN5i-3–CSN5–N8 structure, CSN5i-3–CSN and N8 from the PDB model 1XT9 were fitted into the EM maps and adjusted in Coot. The well-resolved density allowed us to model CSN5i-3 and the zinc ions. For the other CSN[DM]–N8-CRL complexes, the CSN5i-3–CSN–N8-CRL1 model was fitted into the electron microscopy map and adjusted in Coot. For the rest of the CRLs, the N8-CRL structures were automatically built using ModelAngelo[41], combined with the docked CSN[DM] model then further refined manually in Coot. The final model was subjected to real space refinement in Phenix[42]. The iso-peptide bond between the N8 C-terminal glycine residue and the neddylation site lysine of CRLs was manually built and refined with Phenix. The structural figure panels were prepared with PyMol and UCSF ChimeraX.

### DSSO cross-linking of the CSN–N8-CR1 complex

To assemble the CSN–N8-CRL1 complex in vitro for cross-linking, 7 µl of the purified CSN (6.8 µM) and 4.8 µl of N8-CRL1 (30 µM) complexes were mixed and incubated on ice for 10 s or 30 min. For the treated samples, CSN was first incubated with CSN5i-3 at a molar ratio of 1:10 for 30 min on ice before mixing with N8-CRL1. After incubation, the complexes were cross-linked with DSSO at a molar ratio of 1:100 (protein to linker) for 1 h at room temperature[43]. After quenching with 50 mM $NH_4HCO_3$ for 15 min, the cross-linked CSN–N8-CRL1 complexes were transferred onto a 30 kDa microcon and washed with 25 mM $NH_4HCO_3$. The proteins were reduced by 3.3 mM TCEP for 30 min and alkylated with 16.5 mM iodoacetamide for 30 min in dark. After washing with 25 mM $NH_4HCO_3$, proteins were reconstituted in 60 µl of 8 M urea in 25 mM $NH_4HCO_3$. Lys-C (enzyme to protein ratio of 1:100) was added to the solution and the mixture was incubated at 37 °C for 4 h. Urea was then reduced to 1.5 M for trypsin digestion (enzyme to protein ration of 1:50) at 37 °C overnight. The peptide digests were extracted, desalted by Sep-Pak C18 cartridge (Waters) and stored at –80 °C before TMT labelling.

### TMT labelling and SEC-HpHt fractionation

Equal amounts of cross-linked peptides from each condition were labelled with 10-plex TMT reagents (Thermo Fisher Scientific) according to manufacturer's protocol. Briefly, peptides (14.5 µg) were dissolved in 25 µl of 100 mM TEAB (pH 8.5). TMT reagents (0.8 mg) were dissolved in 41 µl of anhydrous acetonitrile (Sigma), and 3 µl of each TMT reagent was added to the corresponding aliquot of peptides together with 7 µl of anhydrous ACN. The reaction was incubated for 1 h with shaking and quenched with 5% hydroxylamine for 15 min at room temperature. The labelled peptides were pooled and dried using Speed-Vac. The dried samples were resuspended in 0.1% TFA buffer and desalted using Sep-Pak C18 cartridge and dried.

The TMT labelled peptides were dissolved in 30% ACN/0.1% TFA, mixed and fractionated on a Superdex 30 Increase 3.2/300 column with an Agilent 1260 Series HPLC. The two fractions containing inter-linked peptides (F23–25min, F25–27min) were collected for direct LC-MS[n] analysis or further HpHt separation[44]. For HpHt separation, each SEC fraction was dissolved in 160 µl of ammonia water (pH 10) and loaded onto a HpHt tip. Peptides were eluted with increasing percentage of ACN in ammonia water (6%, 9%, 12%, 15%, 18%, 21%, 25%, 30%, 35%, and 50%). The 25%, 30%, 35% and 50% fractions were combined to 6%, 9%, 12% and 21% fractions, respectively for LC MS[n] analysis.

### LC MS[n] analysis

Cross-linked peptides were analysed by LC MS[n] using an UltiMate 3000 RSLC coupled with an Orbitrap Fusion Lumos mass spectrometer. Samples were loaded onto a 50 cm × 75 µm Acclaim PepMap C18 column and separated over a 180 min gradient of 4% to 25% acetonitrile at a flow rate of 300 nl min$^{-1}$ (Jiao, 2022). Ions with charge of 4+ to 8+ in the MS[1] scan were selected for MS[2] analysis. The top 4 most abundant fragment ions in MS[2] scan were selected for MS[3] sequencing. The CID-MS[2] normalized collision energy was 23%. For MS[3] scans, CID was used with a collision energy of 35%. TMT quantitation on cross-linked peptides was accomplished using the Synchronous Precursor Selection MS[3] method[28].

### Identification and quantitation of cross-linked peptides

MS[3] spectra was extracted by PAVA (UCSF) and subjected to Protein Prospector (v.6.3.5) for database searching (using Batch-Tag against SwissProt human database, v.2024.08.05, 20,436 entries). The mass tolerances were set as ±20 ppm for parent ions and 0.6 Da for fragment ions. Trypsin was set as the enzyme with a maximum of three missed cleavages allowed. Cysteine carbamidomethylation and TMTplex at protein N-terminus were set as static modifications. A maximum of four variable modifications were also allowed: TMTplex at lysine, methionine oxidation, N-terminal acetylation and N-terminal conversion of glutamine to pyroglutamic acid. Three defined DSSO cross-linked modifications on uncleaved lysines—alkene ($C_3H_2O$, +54 Da), thiol ($C_3H_2SO$, +86 Da) and sulfenic acid ($C_3H_4O_2S$, +104 Da)—were also selected as variable modifications. The in-house software XL-Tools was used to automatically identify, summarize and validate cross-linked peptides based on Protein Prospector database search results and MS[n] data[43]. Quantification values for cross-linked peptides were obtained from reporter ion intensities in Synchronous Precursor Selection MS[3] scans[28]. These intensities were corrected for isotopic impurities of the different TMTplex reagents on the basis of the manufacturer's specifications.

### Affinity purification of CSN complexes

HEK293T cells expressing HBTH-tagged CSN2 and CSN6 were grown to about 90% confluence in DMEM medium containing 10% FBS and 1% Pen/strep[32]. The cells were treated with 1 µM CSN5i-3 or DMSO for 16 h. In-cell cross-linking was then performed using 0.025% formaldehyde for 10 min at 37 °C in PBS buffer[31]. The cells were then washed with PBS and lysed in a native lysis buffer containing 150 mM sodium chloride, 50 mM sodium phosphate, 10% glycerol, 1 mM ATP, 1 mM DTT, 5 mM $MgCl_2$, 1X protease inhibiteo, 1X phosphatase inhibitor and 0.5% NP-40 at pH 7.5. The lysates were centrifuged at 13,000 r.p.m. for 15 min to remove cell debris, and the supernatant was incubated with streptavidin resin for 2 h at 4 °C. The bead-bound proteins were washed with 50 volumes of lysis buffer followed by 20 volumes of another buffer containing 150 mM NaCl, 25 mM $Na_2HPO_4$ and 5% glycerol, pH 7.6, and then cross-linked with 0.5 mM DSSO for 1 h at 37 °C. The cross-linked CSN complexes were reduced and alkylated then digested with LyC/Trypsin. The digested peptides were desalted with C18 tips before LC MS/MS.

### Digestion of cell lysates

To examine protein abundance changes in cells during CSN5i-3 treatment, we digested 1 mg cell lysates from treated and untreated cells using a FASP protocol. Proteins were reduced, alkylated and digested on a 30 kDa microcon (Millipore). The digested peptides were desalted with Sep-Pak column before LC MS/MS.

### LC MS/MS analysis for protein identification and label-free quantitation

For all LC-MS acquisitions, HPLC separation was performed on an UltiMate 3000 UHPLC (Thermo Fisher Scientific) using an EasySpray 50 cm × 75 µm I.D. Acclaim PepMap RSLC column heated at 45 °C and coupled on-line to an Orbitrap Fusion Lumos mass spectrometer (Thermo Fisher Scientific). Specifically, a 4–22% Buffer B gradient (Buffer A, 0.1% formic acid; Buffer B, 100% acetonitrile containing 0.1% formic acid) was run either over 57 min (1.5 h method) or 87 min (2 h method).

## DDA-based analysis

The DDA-MS method for affinity-purified CSN complexes was 90 min and comprised an $MS^1$ survey scan (375–1,800 $m/z$, 120,000 resolution at 200 $m/z$, automatic gain control (AGC) target = $4.0 \times 10^5$, maximum injection time = 50 ms) followed by data-dependent MS/MS acquired in the linear ion trap with HCD NCE30 at top speed for 3 s. Target ions already selected for MS/MS were dynamically excluded for 30 s. $MS^2$ scans were acquired in the linear ion trap using 'Rapid' mode with an AGC target of $2.0 \times 10^4$ and a maximum injection time of 35 ms. Protein quantitation was performed using MaxQuant (v.2.0.3.0) against human proteome sequences from SwissProt database (20,418 entries; March 2025 version)[45]. The first search peptide tolerance was set to 15 ppm, with main search peptide tolerance set to 4.5 ppm. Both peptide spectrum match and protein false discovery rates were set at 1%, in razor peptide fashion. Trypsin was selected for the protease with up to one missed cleavage; no nonspecific cleavage was allowed. For protein quantitation, cysteine carbamidomethylation was set as a fixed modification, while methionine oxidation and N-terminal acetylation were selected for variable modifications, maximum of one per peptide. Intensities were determined as the full peak volume over the retention time profile. Intensities of different isotopic peaks in an isotope pattern were always summed up for further analysis. 'Unique plus razor peptides' was selected as the degree of uniqueness required for peptides to be included in quantification.

## DIA-based analysis

The DIA-MS method for affinity-purified CSN complexes was 90 min and consisted of an $MS^1$ survey scan followed by 33 $MS^2$ scans using variable windows[46]. The $MS^1$ scan range was set as 350–1,650 $m/z$ with a resolution of 120,000 at 200 $m/z$. The AGC target value for the $MS^1$ scan was set to $2.0 \times 10^6$ with a maximum injection time of 50 ms. $MS^2$ scans were acquired at a resolution of 30,000 (at 200 $m/z$), using HCD NCE30. DIA-NN (v.2.0.2) was used for all DIA analysis, including library generation, protein identification and quantitation[47]. The spectral library was generated using human proteome sequences from SwissProt database (20,418 entries; March 2025 version). Trypsin was set as the protease with one missed cleavage. N-term excision and cysteine carbamidomethylation were set as fixed modifications, whereas methionine oxidation was set as a variable modification with one max per peptide. The peptide length was set to 7–30 aa, with charge 2–4. The precursor false discovery rate was set to 1% for the output and proteins were quantified using the QuantUMS strategy[48].

## PRM-based targeted quantitative analysis

To generate the PRM target library for quantitation, peptide digests from affinity-purified CSN complexes were pooled and run using 2 h DIA-MS methods with the same settings as outlined above. The top 3 abundant peptide ions from the selected 126 proteins were chosen, with preference towards unmodified peptides and sequences identified without missed cleavages where possible. A total of 351 precursors from 126 proteins were selected and monitored across two 2 h acquisitions for cell lysate samples, with each peptide being scheduled within a 4 min window. The isolation window was set as 0.7 Da and Orbitrap resolution was 30,000 at 200 $m/z$. Transitions determined by DIA-MS analysis were quantified using Skyline (v.23.1.0.268)[49]. The peak areas of all transitions were summed for each peptide and then averaged to represent protein abundances. Final protein abundances were then averaged across biological replicates and compared between treated and control lysates. Two biological replicates were performed for each condition.

## Validation of the selected CSN interactions by immunoblotting analysis

Affinity-purified human CSN complexes were separated by SDS–PAGE to validate the selected CSN interactions. Proteins were transferred to a PVDF membrane and analysed by immunoblotting. HBTH-CSN2 and HBTH-CSN6 were detected using a streptavidin–HRP conjugate (1:10,000), DDB2 was detected by using anti-DDB2 antibody (Thermo Fisher Scientific MA5-34832, 1:500 dilution), CUL4A was detected by using anti-CUL4A (Thermo Fisher Scientific PA5-14542, 1:1,000 dilution) and FBXO22 was detected by using anti-FBXO22 (Proteintech 13606-1-AP, 1:500 dilution).

## Affinity pull-down analysis

Approximately 200 μg of purified CSN complex with His-Venus-tagged CSN5 was used as the bait. Venus-tagged N8 (wild-type and three mutants) were added at a 25:1 molar excess relative to CSN, either in the presence or absence of 10 μM CSN5i-3. The bait and binding partners were incubated together on ice at 4 °C for 2 h with gentle mixing. The reaction mixtures were then applied to 50 μl of Ni–NTA Sepharose resin and incubated with end-over-end rotation at 4 °C for 3 h. Beads were washed extensively with a binding buffer containing 20 mM Tris-HCl, pH 8.0, 150 mM NaCl and 10 mM imidazole. Bound complexes were eluted in 100 μl of an elution buffer containing 20 mM Tris-HCl, pH 8.0, 150 mM NaCl and 300 mM imidazole. Eluates were resolved by SDS–PAGE and the Venus signal was detected by in-gel fluorescence imaging.

## Inhibition of iso-peptidase activity by CSN5i-3 and CSN5i-1a

The enzymatic activities of CSN and CSN5–CSN6 were determined by monitoring the amounts of the deneddylation product, N8 conjugated with the fluorescent dye CF633 and the reaction substrate, CF633–N8-CRL1. Twenty-microlitre reactions were performed in PCR tubes. Enzymes (recombinant CSN or CSN5–CSN6) were used at 2.0 nM (added at $T = 0$), and CF633–N8-CRL1 was used at 2 μM. CSN5i-3 was serially diluted threefold from 450 nM down to 0.062 nM. CSN5i-1a was serially diluted threefold from 100 μM down to 137 nM. Reactions at room temperature were stopped after 25 min by adding 8 μl of 4X SDS loading buffer containing Orange G, then heated for 5 min at 90 °C. Two tubes without inhibitor were designated as 100% activity, and one tube that contained SDS loading buffer before enzyme addition was considered 0% activity. The reaction products N8–CF633 and the CF633–N8-CRL1 substrate were separated on 4–15% mini-Protein gels (BioRad) at 250 V (67 mA) for 20–30 min. Fluorescence signals from CF633 were detected using an Odyssey CLX Imaging System (LICOR) in the 700 nm channel. All fluorescence signals were normalized to correct for loading variations. The Michaelis constant (Km) was determined for both CSN and CSN5–CSN6 with the CF633–N8-CRL1 substrate under the same buffer and reaction conditions. The substrate was serially diluted threefold from 1 μM to 0.021 nM. Fluorescence signals from CF633 were detected using the Odyssey CLX Imaging System (LICOR) in the 700 nm channel, and the enzyme rate versus substrate concentration was analysed using an enzymatic kinetic model in Prism (GraphPad).

## Reporting summary

Further information on research design is available in the Nature Portfolio Reporting Summary linked to this article.

## Data availability

The coordinates and cryo-EM maps have been deposited in the PDB and the Electron Microscopy Data Bank with the following accession codes: CSN, 9E5Z, EMD-47532; CSN5i-3–CSN, 9E81, EMD-47698; CSN5i-1a-CSN, 9PH4, EMD-71639; CSN5i-3–CSN–N8-CRL1, 9EFV, EMD-47981, EMD-47729 and EMD-47767; CSN5i-3–CSN-N8, 9E77 and EMD-47660; CSN[DM]–N8-CRL1, 9EFM, EMD-47976, EMD-47500 and EMD-47502; CSN[DM]–N8-CRL2, 9EFQ, EMD-47977, EMD-47701 and EMD-47702; CSN[DM]–N8-CRL3, 9EGL, EMD-47990, EMD-47776 and EMD-47985; CSN[DM]–N8-CRL4A, 9EG8, EMD-47986, EMD-47543 and EMD-47699; CSN[DM]–N8-CRL5, 9EG1, EMD-47983, EMD-47663 and EMD-47665. The mass spectrometry data associated with XL-MS and AP-MS have been

deposited to the ProteomeXchange Consortium via the PRIDE partner repository with the dataset identifier PXD063786 and PXD063822, respectively.

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

**Acknowledgements** We thank R. Yan, X. Zhao, J. Jung, Z. Yu, J. D. Quispe and S. Dickinson for assistance with electron microscopy data acquisition, and P. Morenkov for assistance with formatting TMT XL-MS data. N.Z. is a Howard Hughes Medical Institute Investigator. This work is supported by the NIH (grant nos. R35GM145249 and R01CA290875 to L.H., R01AI179768 and R01AI152358 to E.F.) and ISF (grant no. 737/23 to D.S.).

**Author contributions** H.S., H.M., L.H. and N.Z. conceived the project. H.M. and H.S. constructed and purified all proteins samples in this study with the help from S.C. H.S. performed all cryo-EM grid preparation, specimen screening, data collection and processing tasks. T.R.H. and H.S. performed the in vitro enzymatic and binding assays. X.W. prepared samples for mass spectrometry analyses and performed biochemical experiments. C.Y. performed mass spectrometry experiments and data analysis. F.J. performed in vitro cross-linking and mass spectrometry experiments. M.B., B.S. and D.S. analysed the in vitro cross-linking mass spectrometry data. Z.Z. and E.F. synthesized CSN5i-1a. The manuscript was written by N.Z., H.S. and L.H. with inputs from all authors.

**Competing interests** N.Z. is one of the scientific co-founders of SEED Therapeutics, and also serves as a member of the scientific advisory board of Synthex, Molecular Glue Labs, Differentiated Therapeutics and Cold Start Therapeutics. The other authors declare no competing interests.

**Additional information**
**Correspondence and requests for materials** should be addressed to Lan Huang or Ning Zheng.

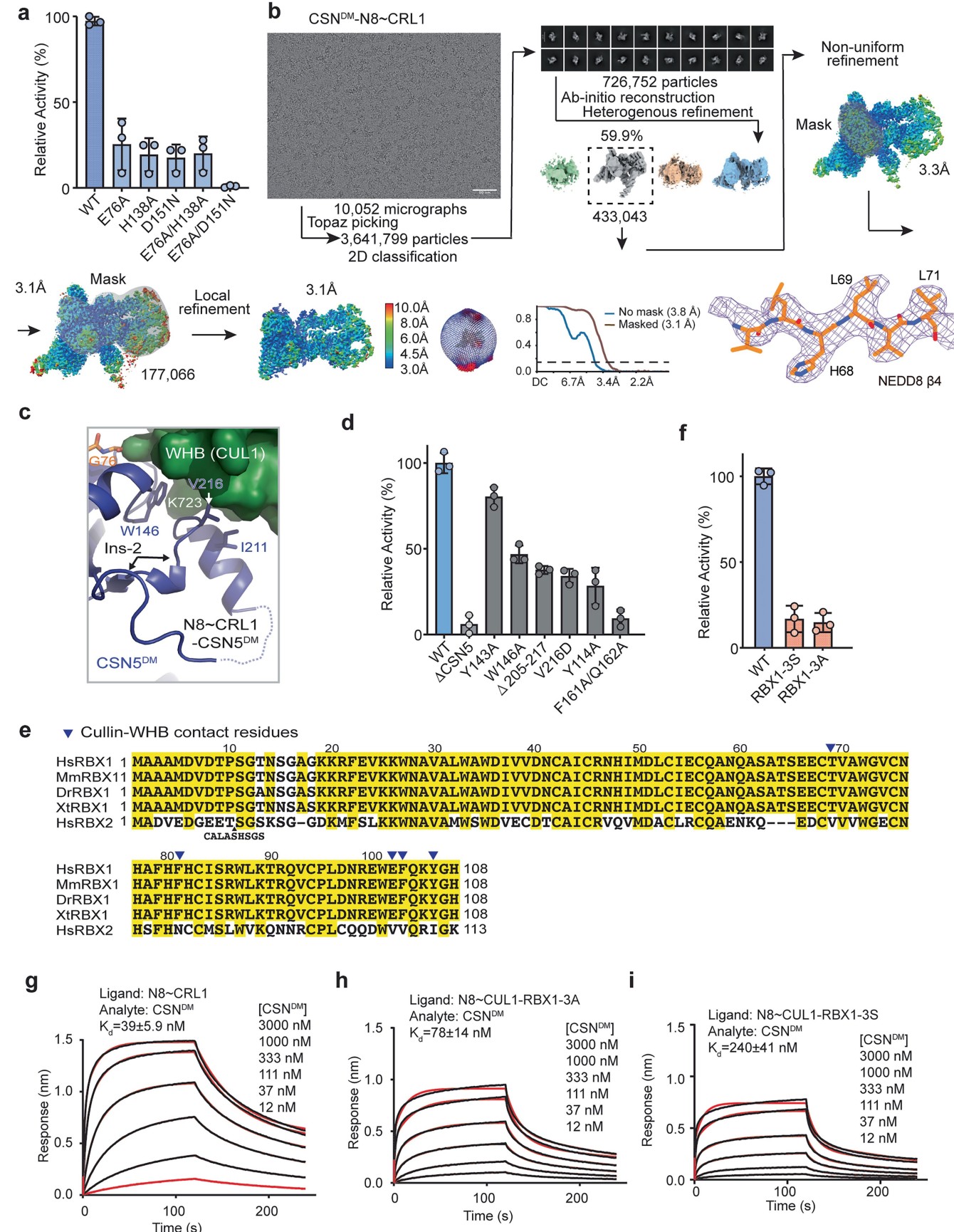

**Extended Data Fig. 1** | See next page for caption.

**Extended Data Fig. 1 | Structure determination of the pre-catalytic state of CSN. a**. A comparison of the enzymatic activities of CSN containing wild type (WT) and mutant CSN5 catalytic subunit. Data are mean ± s.d. of n = 3 biological replicates. **b**. Cryo-EM data processing for the CSN$^{DM}$-N8 - CRL1 complex. **c**. A close-up view of the CSN5 Ins-2 loop in the CSN$^{DM}$-N8 - CRL1 complex. **d**. A comparison of the enzymatic activities of CSN containing WT and mutant CSN5 designed for structure validation. Data are mean ± s.d. of n = 3 biological replicates. **e**. Sequence alignment of RBX1 orthologues and human RBX2.

Hs: *Homo sapiens*, Mm: *Mus musculus*, Dr: *Danio rerio*, Xt: *Xenopus tropical*. Cullin-WHB-contacting residues are indicated by blue triangles. **f**. A comparison of the enzymatic activities of CSN towards N8 - CRL1 featuring WT and mutant RBX1 with Phe81, Phe103, and Y106 simultaneously mutated to serine or alanine. Data are mean ± s.d. of n = 3 biological replicates. **g, h, i**. Affinity measurement of CSN$^{DM}$ to N8 - CRL1 containing wild type and mutant RBX1 by BLI. RBX1-3A and RBX1-3S feature the three aromatic residues at the RBX1-WHB interface mutated to alanine and serine, respectively.

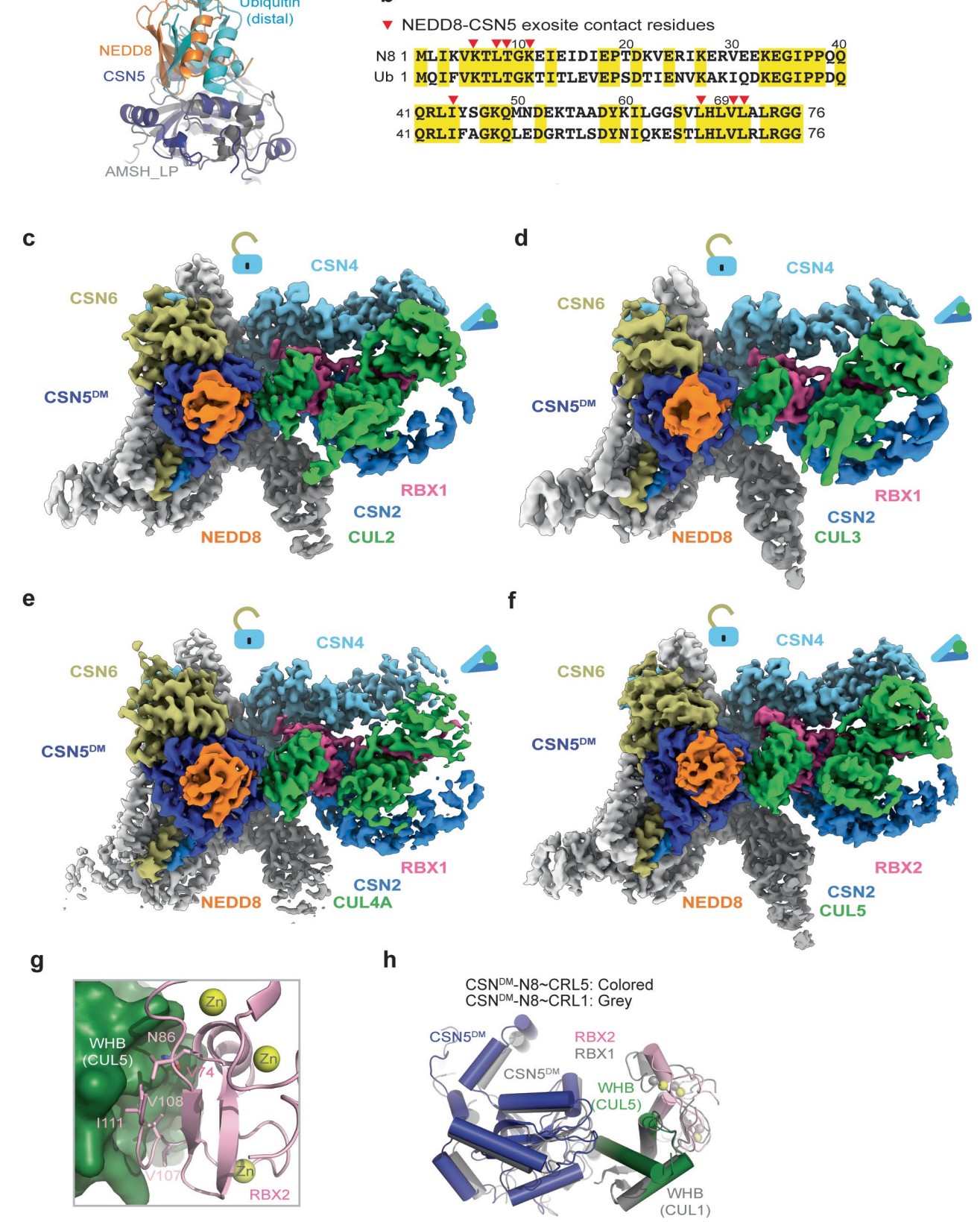

**Extended Data Fig. 2 | Structures of the pre-catalytic state of CSN in complexes with different N8–CRLs. a**. A comparison of the binding mode of N8 to CSN5 in the CSN5i-3–CSN–N8-CRL1 complex to that of the distal ubiquitin to AMSH-LP in the AMSH-LP K63-linked ubiquitin dimer complex (PDB:2ZNV). **b**. Sequence alignment of human N8 and ubiquitin. CSN5-contacting residues in N8 at the N8-CSN5 exosite are indicated by red triangles. Conversed residues are highlighted in yellow. **c**–**f**. Cryo-EM maps of CSN bound to N8-CRL2, N8-CRL3, N8-CRL4A, and N8-CRL5. **g**. A close-up view of the interface between CUL5-WHB and RBX2-RING. **h**. A slight positional shift of CSN5 in the structure of CSN$^{DM}$ in complexes with N8-CRL1 versus N8-CRL5.

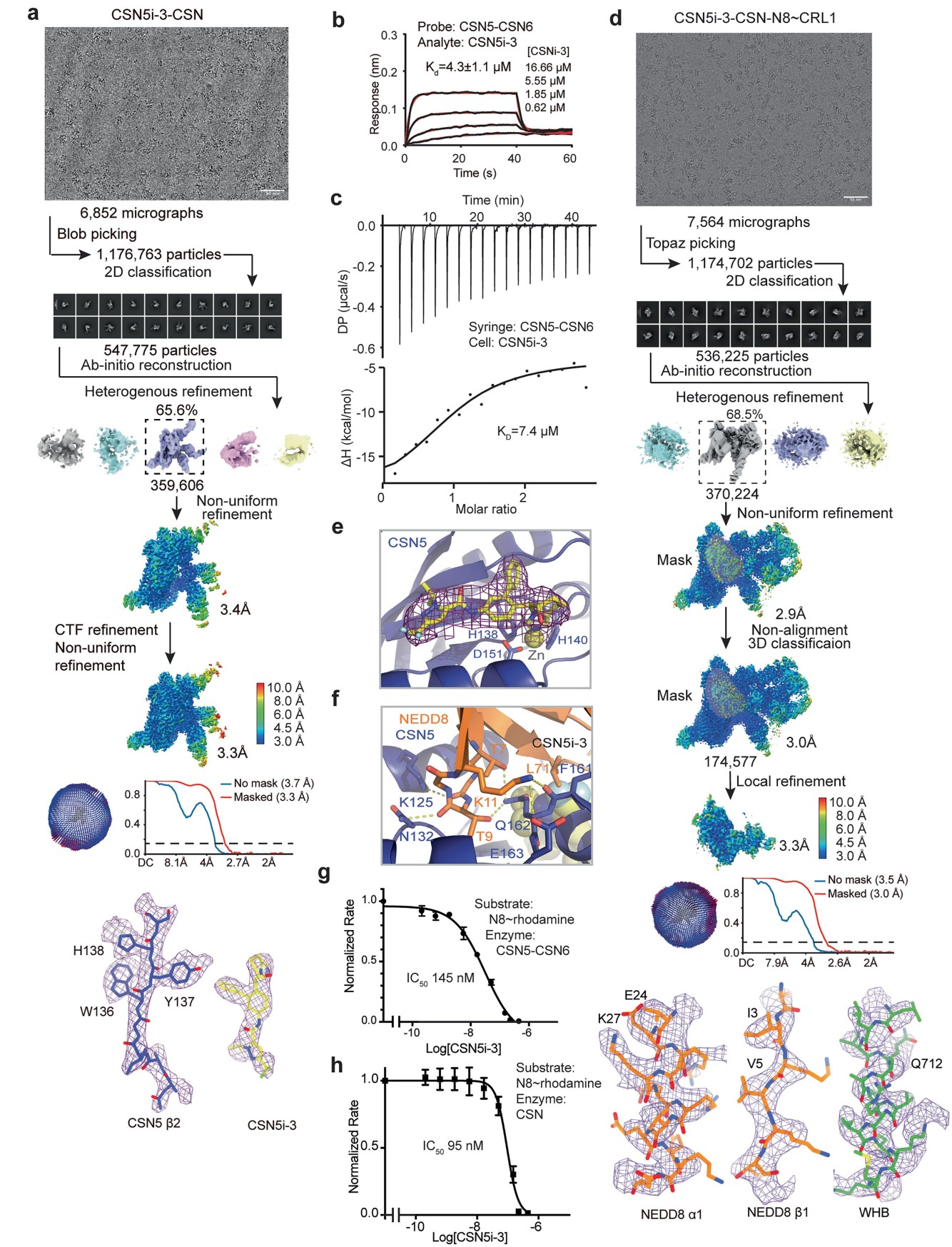

**Extended Data Fig. 3** | See next page for caption.

**Extended Data Fig. 3 | CSN5i-3 as an orthosteric inhibitor of CSN. a**. Cryo-EM data process of the CSN5i-3–CSN complex. **b**. Binding affinity of CSN5i-3 to the CSN5–CSN6 heterodimer determined by BLI. **c**. Binding affinity of CSN5i-3 to the CSN5–CSN6 heterodimer determined by ITC. **d**. Cryo-EM data process of the CSN5i-3–CSN–N8 - CRL1 complex. **e**. A close-up view of the CSN5 catalytic site occupied by CSN5i-3 with its density revealed in the cryo-EM map of the CSN5i-3–CSN complex. **f**. A close-up view of the interface between CSN5 and the tip of the β1-β2 loop of N8. Hydrogen bonds and salt bridges are shown in yellow dash lines. The side chains of interfacial residues are shown in stick. The nearby CSN5i-3 is shown in the background. **g**. The potency of CSN5i-3 in inhibiting the iso-peptide activity of the CSN5–CSN6 heterodimer towards N8-Rhodamine. Data are mean ± s.d. of n = 3 biological replicates. **h**. The potency CSN5i-3 in inhibiting the iso-peptide activity of intact CSN5 towards N8-Rhodamine. Data are mean ± s.d. of n = 3 biological replicates.

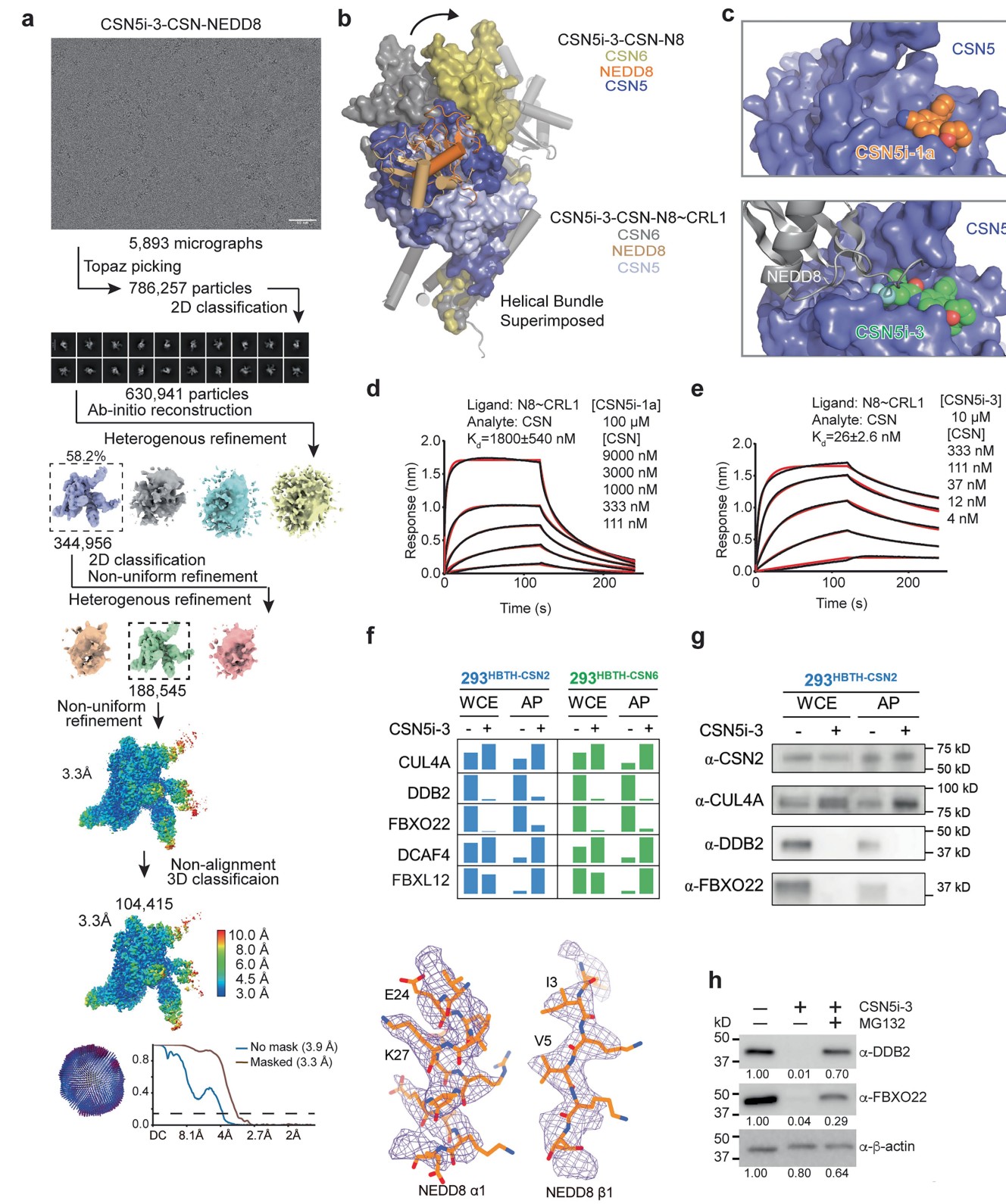

**Extended Data Fig. 4 | CSN5i-3 as an orthosteric molecular glue. a**. Cryo-EM data process of the CSN5i-3–CSN-N8 complex. **b**. Orientational difference of the CSN5–CSN6 dimer within CSN between the CSN5i-3–CSN–N8 - CRL1 and CSN5i-3–CSN–N8 complex structures. **c**. A comparison between the binding mode of CSN5i-1a (top panel) and CSN5i-3 (bottom panel) to CSN. Both compounds are shown in spheres. CSN5 is shown in surface representation. N8 bound to CSN5i-3-occupied CSN is shown in grey cartoon. **d**. The affinity of CSN towards N8-CRL1 in a saturating amount of CSN5i-1a measured by BLI. **e**. The affinity of CSN towards N8-CRL1 in a saturating amount of CSN5i-3

measured by BLI. **f**. Abundances of CUL4A, DDB1, FBXO22, DCAF4 and FBXL12 in WCE and affinity-purified (AP) CSN2 and CSN6 complexes determined by quantitative MS analyses. For gel source data, see Supplementary Fig. 1. **g**. Assessment of CUL4A, DDB1, and FBXO22 abundances in WCE and affinity-purified CSN2 complexes in the absence and presence of CSN5i-3 using immunoblotting analysis. The experiment has been repeated three times with similar results. **h**. Proteasomal degradation of DDB2 and FBXO22 induced by CSN5i-3. The experiment has been repeated three times with similar results. For gel source data, see Supplementary Fig. 1.

**Extended Data Table 1 | Cryo-EM image processing and model refinement statistics of CSN in complex with N8~CRLs**

| | CSN$^{DM}$-N8~CRL1 (EMDB EMD-47500, EMD-47502, EMD-47976) (PDB 9EFM) | CSN$^{DM}$-N8~CRL2 (EMDB EMD-47701, EMD-47702, EMD-47977) (PDB 9EFQ) | CSN$^{DM}$-N8~CRL3 (EMDB EMD-47776, EMD-47985, EMD-47990) (PDB 9EGL) | CSN$^{DM}$-N8~CRL4A (EMDB EMD-47543, EMD-47699, EMD-47986) (PDB 9EG8) | CSN$^{DM}$-N8~CRL5 (EMDB EMD-47663, EMD-47665, EMD-47983) (PDB 9EG1) |
|---|---|---|---|---|---|
| **Data collection and processing** | | | | | |
| Magnification Voltage (kV) | 300 | 300 | 200 | 300 | 300 |
| Electron exposure (e–/Å$^2$) | 60 | 60 | 50 | 60 | 60 |
| Defocus range (μm) | 0.8~3 | 0.8~2.5 | 0.8~2.5 | 0.8~3 | 0.8~3 |
| Pixel size (Å) | 0.839 | 0.852 | 0.885 | 0.84 | 0.835 |
| Symmetry imposed | C1 | C1 | C1 | C1 | C1 |
| Initial particle images (no.) | 3,641,799 | 2,468,106 | 933,332 | 1,873,690 | 3429,756 |
| Final particle images (no.) | 177,066 | 205,270 | 308,367 | 163,795 | 239,911 |
| Map resolution (Å) FSC threshold | 3.2 | 3.0 | 3.9 | 3.4 | 3.5 |
| Map resolution range (Å) | 3.2~10 | 3.0~10 | 3.9~10 | 3.4~10 | 3.5~10 |
| | | | | | |
| **Refinement** | | | | | |
| Initial model used (PDB code) | 4D10, 1LDJ | 4D10, 8R5H | 4D10, 8K9I | 4D10, 2HYE | 4D10, 6V9I |
| Model resolution (Å) FSC threshold | 3.16 0.143 | 2.96 0.143 | 3.93 0.143 | 3.39 0.143 | 3.52 0.143 |
| Map sharpening $B$ factor (Å$^2$) | -87.7 | -76.4 | -124.7 | -101.6 | -102.6 |
| Model composition Non-hydrogen atoms Protein residues Ligands | 24934 3105 ZN:3 | 25025 3119 ZN: 3 | 25447 3169 ZN: 3 | 25410 3162 ZN: 3 | 23719 3067 ZN: 3 |
| $B$ factors (Å$^2$) Protein Ligand | 60.89 60.82 | 21.45 17.43 | 161.49 269.80 | 88.72 66.90 | 117.74 89.3 |
| R.m.s. deviations Bond lengths (Å) Bond angles (°) | 0.004 (2) 0.647 (16) | 0.005 (0) 0.666 (4) | 0.003 (0) 0.667 (7) | 0.003 (0) 0.620 (8) | 0.003 (0) 0.755 (24) |
| Validation MolProbity score Clashscore Poor rotamers (%) | 1.86 8.45 0.33 | 2.02 9.02 1.68 | 1.97 10.26 0.00 | 1.97 8.43 0.38 | 1.85 7.93 0.94 |
| Ramachandran plot Favored (%) Allowed (%) Disallowed (%) | 94.04 5.73 0.23 | 94.58 5.19 0.23 | 93.28 6.24 0.48 | 94.95 4.67 0.38 | 93.73 5.61 0.66 |

**Extended Data Table 2 | Cryo-EM image processing and model refinement statistics of CSN5i-3 bound CSN, CSN-N8, and CSN–N8~CRL1**

| | CSN (EMDB EMD-47532) (PDB 9E5Z) | CSN5i-3-CSN (EMDB EMD-47698) (PDB 9E81) | CSN5i-3-N8-CSN (EMDB EMD-47660) (PDB 9E77) | CSN5i-3-CSN-N8~CRL1 (EMDB EMD-47729, EMD-47767, EMD-47981) (PDB 9EFV) | CSN5i-1a-CSN (EMDB EMD-71639) (PDB 9PH4) |
|---|---|---|---|---|---|
| **Data collection and processing** | | | | | |
| Magnification Voltage (kV) | 300 | 200 | 300 | 300 | 300 |
| Electron exposure (e–/Å²) | 60 | 50 | 60 | 60 | 60 |
| Defocus range (μm) | 0.8~3 | 0.8~2.5 | 0.8~3 | 0.8~3 | 0.8~3 |
| Pixel size (Å) | 0.84 | 0.885 | 0.84 | 0.827 | 0.826 |
| Symmetry imposed | C1 | C1 | C1 | C1 | C1 |
| Initial particle images (no.) | 1,412,329 | 1,176,763 | 786,257 | 1,174,702 | 1,504,944 |
| Final particle images (no.) | 280,410 | 359,606 | 188,545 | 174,577 | 451,908 |
| Map resolution (Å) FSC threshold | 3.3 | 3.3 | 3.3 | 3.0 | 3.0 |
| Map resolution range (Å) | 3.3~9 | 3.3~10 | 3.3~10 | 3.0~10 | 3.0~10 |
| | | | | | |
| **Refinement** | | | | | |
| Initial model used (PDB code) | 4D10 | 4D10, 5JOG | 4D10, 5JOG | 4D10, 1LDJ, 5JOG | 4D10, 5JOH |
| Model resolution (Å) FSC threshold | 3.30 0.143 | 3.29 0.143 | 3.26 0.143 | 3.03 0.143 | 2.95 0.143 |
| Map sharpening $B$ factor (Å²) | -130.3 | -95.6 | -92.1 | -73 | -96.6 |
| Model composition Non-hydrogen atoms Protein residues Ligands | 18279 2285 ZN:1 | 17913 2236 ZN:1 6LT:1 | 18626 2326 ZN:1 6LT:1 | 23719 3067 ZN:4 6LT:1 | 15982 1996 ZN:1 A1CH3:1 |
| $B$ factors (Å²) Protein Ligand | 74.10 68.74 | 37.93 16.77 | 33.00 14.93 | 49.55 29.44 | 59.46 70.44 |
| R.m.s. deviations Bond lengths (Å) Bond angles (°) | 0.003 (0) 0.589 (25) | 0.003 (0) 0.553 (5) | 0.003 (3) 0.570 (15) | 0.003 (0) 0.658 (16) | 0.003 (0) 0.581 (8) |
| Validation MolProbity score Clashscore Poor rotamers (%) | 1.77 6.04 0.15 | 1.74 6.30 0.26 | 1.74 7.24 0.20 | 1.93 9.43 0.37 | 2.02 6.71 3.04 |
| Ramachandran plot Favored (%) Allowed (%) Disallowed (%) | 93.19 6.55 0.27 | 94.25 5.66 0.09 | 95.12 4.75 0.13 | 93.50 6.10 0.39 | 95.94 3.90 0.15 |

# Reporting Summary

## Statistics

For all statistical analyses, confirm that the following items are present in the figure legend, table legend, main text, or Methods section.

| n/a | Confirmed | |
|---|---|---|
| ☐ | ☒ | The exact sample size (*n*) for each experimental group/condition, given as a discrete number and unit of measurement |
| ☐ | ☒ | A statement on whether measurements were taken from distinct samples or whether the same sample was measured repeatedly |
| ☒ | ☐ | The statistical test(s) used AND whether they are one- or two-sided *Only common tests should be described solely by name; describe more complex techniques in the Methods section.* |
| ☒ | ☐ | A description of all covariates tested |
| ☒ | ☐ | A description of any assumptions or corrections, such as tests of normality and adjustment for multiple comparisons |
| ☐ | ☒ | A full description of the statistical parameters including central tendency (e.g. means) or other basic estimates (e.g. regression coefficient) AND variation (e.g. standard deviation) or associated estimates of uncertainty (e.g. confidence intervals) |
| ☒ | ☐ | For null hypothesis testing, the test statistic (e.g. *F*, *t*, *r*) with confidence intervals, effect sizes, degrees of freedom and *P* value noted *Give P values as exact values whenever suitable.* |
| ☒ | ☐ | For Bayesian analysis, information on the choice of priors and Markov chain Monte Carlo settings |
| ☒ | ☐ | For hierarchical and complex designs, identification of the appropriate level for tests and full reporting of outcomes |
| ☒ | ☐ | Estimates of effect sizes (e.g. Cohen's *d*, Pearson's *r*), indicating how they were calculated |

*Our web collection on statistics for biologists contains articles on many of the points above.*

## Software and code

Policy information about availability of computer code

| Data collection | Cryo-EM: Krios and Glacios Transmission Electron Microscope (Thermo Fisher) with K3 direct electron detector, operated on SerialEM (v4.1). Mass spectrometry data: collected with Orbitrap Fusion Lumos Tune Application (v3.5.3890). BLI data were collected with Octet BLI Discovery (v13.0.3.26). ITC data were collected with MicroCal PEAQ-ITC Control Software (v1.41). |
|---|---|
| Data analysis | Cryo-EM datasets were processed with cryoSPARC (v4.4.1). The model building used crystal structure model as template, then docked in ChimeraX-1.6 or automatically built with ModelAngelo (v1.0), adjusted with Coot (v0.9.8.91) and refined with PHENIX (v1.20.1-4487-000). Mass spectrometry data were searched using Protein Prospector (v6.3.5; http://prospector.ucsf.edu/prospector/mshome.htm), which was developed by UCSF mass spectrometry facility. Cross-linked peptides were identified through the integration of MS1, MS2 and MS3 data using XL-tools scripts (Wang, et al. Mol Cell Proteomics 2017, 16 (5), 840-854)). MaxQuant (v2.0.3.0) was used for DDA-based label-free quantitation, and DIA-NN (v2.0.2) was used for DIA-based quantitation. BLI data was analyzed with Octet BLI Analysis (v12.2). ITC data were analyzed with MicroCal PEAQ-ITC Analysis Software (v1.41). |

For manuscripts utilizing custom algorithms or software that are central to the research but not yet described in published literature, software must be made available to editors and reviewers. We strongly encourage code deposition in a community repository (e.g. GitHub). See the Nature Portfolio guidelines for submitting code & software for further information.

# Data

Policy information about availability of data

All manuscripts must include a data availability statement. This statement should provide the following information, where applicable:

- Accession codes, unique identifiers, or web links for publicly available datasets
- A description of any restrictions on data availability
- For clinical datasets or third party data, please ensure that the statement adheres to our policy

The coordinates and cryo-EM maps were deposited in the Protein Data Bank (PDB) and the Electron Microscopy Data Bank (EMDB) with the following accession numbers: CSN: 9E5Z, EMD-47532; CSN5i-3-CSN: 9E81, EMD-47698; CSN5i-1a-CSN: 9PH4, EMD-71639; CSN5i-3-CSN-N8~CRL1: 9EFV, EMD-47981, EMD-47729, EMD-47767; -CSN5i-3-CSN-NEDD8: 9E77, EMD-47660; CSNDM-N8~CRL1: 9EFM, EMD-47976, EMD-47500, EMD-47502; CSNDM-N8~CRL2: 9EFQ, EMD-47977, EMD-47701, EMD-47702; CSNDM-N8~CRL3: 9EGL, EMD-47990, EMD-47776, EMD-47985; CSNDM-N8~CRL4A: 9EG8, EMD-47986, EMD-47543, EMD-47699, and CSNDM-N8~CRL5: 9EG1, EMD-47983, EMD-47663, EMD-47665.
The mass spectrometry data for cross-linking and affinity pulldown [HS15.1]have been deposited to the ProteomeXchange Consortium via the PRIDE partner repository with the dataset identifier PXD063786, PXD063822, respectively.

# Research involving human participants, their data, or biological material

Policy information about studies with human participants or human data. See also policy information about sex, gender (identity/presentation), and sexual orientation and race, ethnicity and racism.

| | |
|---|---|
| Reporting on sex and gender | N/A |
| Reporting on race, ethnicity, or other socially relevant groupings | N/A |
| Population characteristics | N/A |
| Recruitment | N/A |
| Ethics oversight | N/A |

Note that full information on the approval of the study protocol must also be provided in the manuscript.

# Field-specific reporting

Please select the one below that is the best fit for your research. If you are not sure, read the appropriate sections before making your selection.

☒ Life sciences    ☐ Behavioural & social sciences    ☐ Ecological, evolutionary & environmental sciences

For a reference copy of the document with all sections, see nature.com/documents/nr-reporting-summary-flat.pdf

# Life sciences study design

All studies must disclose on these points even when the disclosure is negative.

| | |
|---|---|
| Sample size | No animal or human samples were used in this study. All experiments were performed exclusively with purified recombinant protein complexes, and cell lysates in standard biochemical and structural assays. Therefore, statistical sample size determination is not applicable. For biochemical experiments, we ensured reproducibility by performing each assay in biological replicates (typically n = 3) and confirming key findings by independent replicates. This approach follows established practice for in-vitro enzymology and protein biochemistry where experimental reproducibility—not population sampling—is the relevant metric. |
| Data exclusions | No data exclusions were performed in this study. |
| Replication | Where indicated in the paper, experiments were performed in replicate (duplicate or triplicate). Replicate type is specified in the text. All attempts at replication were successful. |
| Randomization | Randomization is not applicable to this study. All experiments were performed using purified recombinant proteins, and standard cell culture systems used solely for protein expression. These experimental systems do not involve allocation of subjects, biological groups, or treatment arms. Instead, reproducibility was ensured by performing biochemical and biophysical assays in independent replicates, which is the standard practice for mechanistic protein studies. |
| Blinding | Investigators were not blinded for any of the experiments performed in this study. This was done for practical purposes, and is standard practice for studies employing biochemistry, cell culture, and genomics (Zhang et al., 2018; Vinyard et al., 2019; Haggerty et al., 2021). |

# Reporting for specific materials, systems and methods

We require information from authors about some types of materials, experimental systems and methods used in many studies. Here, indicate whether each material, system or method listed is relevant to your study. If you are not sure if a list item applies to your research, read the appropriate section before selecting a response.

## Materials & experimental systems

| n/a | Involved in the study |
|-----|----------------------|
| ☐ | ☒ Antibodies |
| ☐ | ☒ Eukaryotic cell lines |
| ☒ | ☐ Palaeontology and archaeology |
| ☒ | ☐ Animals and other organisms |
| ☒ | ☐ Clinical data |
| ☒ | ☐ Dual use research of concern |
| ☒ | ☐ Plants |

## Methods

| n/a | Involved in the study |
|-----|----------------------|
| ☒ | ☐ ChIP-seq |
| ☒ | ☐ Flow cytometry |
| ☒ | ☐ MRI-based neuroimaging |

## Antibodies

**Antibodies used**

1) Streptavidin HRP to detect HBTH-CSN2, HBTH-CSN6: Thermo Fisher Scientific, Streptavidin Protein, HRP Cat# PI21126
2) CUL4A antibody: Invitrogen / Thermo Fisher Scientific, Cat# PA5-14542
3) DDB2 antibody: Invitrogen / Thermo Fisher Scientific, Cat# MA5-34832
4) FBXO22 antibody: Proteintech, Cat# 13606-1-AP

**Validation**

Streptavidin–HRP (Thermo Scientific, Streptavidin Protein, HRP, Cat# 21126) was validated by the manufacturer; application data and supporting citations are provided at: https://www.thermofisher.com/order/catalog/product/21126
CUL4A (Invitrogen / Thermo Fisher Scientific, Cat# PA5-14542) was validated by the manufacturer; application data and supporting citations are provided at: https://www.thermofisher.com/antibody/product/Cullin-4A-Antibody-Polyclonal/PA5-14542
DDB2 (Invitrogen / Thermo Fisher Scientific, Cat# MA5-34832) was validated by the manufacturer; application data and supporting citations are provided at: https://www.thermofisher.com/antibody/product/DDB2-Antibody-clone-JE16-41-Recombinant-Monoclonal/MA5-34832
FBXO22 (Proteintech, 13606-1-AP; RRID: AB_2104403) was validated by the manufacturer; application data and supporting citations are provided at: https://www.ptglab.com/products/FBXO22-Antibody-13606-1-AP.htm

## Eukaryotic cell lines

Policy information about cell lines and Sex and Gender in Research

**Cell line source(s)**

HEK293T was obtained from ATCC, Cat# CRL-3216. HEK293 cell lines stably expressing HBTH-tagged CSN2 or HBTH-tagged CSN6. ExpiSF9 cell line was obtained from Thermo Fisher (A35243).

**Authentication**

All cell lines were authenticated by Short Tandem Repeat profiling (Genetica). Stable cell lines were also authenticated using affinity purification and western blot analysis.

**Mycoplasma contamination**

All cell lines tested negative for mycoplasma.

**Commonly misidentified lines**
(See ICLAC register)

No commonly misidentified cell lines were used.

# Plants

**Seed stocks**

*Report on the source of all seed stocks or other plant material used. If applicable, state the seed stock centre and catalogue number. If plant specimens were collected from the field, describe the collection location, date and sampling procedures.*

**Novel plant genotypes**

*Describe the methods by which all novel plant genotypes were produced. This includes those generated by transgenic approaches, gene editing, chemical/radiation-based mutagenesis and hybridization. For transgenic lines, describe the transformation method, the number of independent lines analyzed and the generation upon which experiments were performed. For gene-edited lines, describe the editor used, the endogenous sequence targeted for editing, the targeting guide RNA sequence (if applicable) and how the editor was applied.*

**Authentication**

*Describe any authentication procedures for each seed stock used or novel genotype generated. Describe any experiments used to assess the effect of a mutation and, where applicable, how potential secondary effects (e.g. second site T-DNA insertions, mosiacism, off-target gene editing) were examined.*

