## [Peer Review file · Nature]

CSN5i-3: An Orthosteric Molecular Glue Inhibitor of COP9 Signalosome

Corresponding Author: Professor Ning Zheng

Version 0:

Reviewer comments:

Referee #1

(Remarks to the Author)

Here the authors essentially discover an allosteric mechanism through which orthosteric inhibitors not only block substrate binding at the active site but enhance their potency by allosterically altering the exosite conformation, leading to exposing the surface of the exosite in a binding-favored state, to which the substrate binds. In turn, substrate binding at the exosite cooperatively enhances their inhibitory action. They beautifully show that CSN5i-3, an orthosteric inhibitor of the COP9 signalosome (CSN), which catalyzes NEDD8 deconjugation from the cullin-RING ubiquitin ligases (CRLs), thus promotes the substrate-enzyme assembly, which cooperatively stabilizes its own interaction at the CSN's active site. Whereas the authors highlight it as unlikely, I would not call their observation as such. Allosteric effects are bi-directional. They are cooperative. An allosteric binding partner can modulate – enhance or diminish– the efficacy of the orthosteric ligand (inhibitor), while concomitantly, the orthosteric ligand (inhibitor) can modulate the efficacy of the allosteric ligand.

I find this work exciting not only because of the novelty and its potential impact on the crucial CSN system, but also because of its potential broad implications. Any binding at any site on the surface will stimulate a shift of the protein ensembles. Here the shift elicited by CSN5i-3 leads to conformational changes at the NEDD8-binding exosite, hindering CSN5 cleavage of NEDD8 modification from Cullin-RING E3 ubiquitin ligases (CRLs), thus trapping them in their neddylated, active state. These conformational changes are crucial, as they expose the occluded loop in the CSN holoenzyme structure. At the same time, the shift of the ensemble generated by the allosteric binding of the substrate cooperatively increases the efficacy of CSN5i-3. Thus, they discovered that CSN5i-3 functions as both an orthosteric inhibitor of the iso-peptidase target, and a stabilizer of the enzyme-substrate complex, which led them to coin orthosteric molecular glue (OMG) inhibitors. I would add that its efficacy is further strengthened allosterically.

I have only one comment: CSN does not undergo major conformational changes upon substrate engagement as the paper notes. Instead, the conformational changes were likely promoted by the inhibitor binding and are captured by the substrate. They may or may not be observed in the pre-substrate binding state, depending on the relative stability of the state – stable or sparse. A limited resolution may suggest a broad distribution, that is a sparse state.

Referee #2

(Remarks to the Author)

The manuscript by Shi and colleagues reports structures of CSN in complex with its inhibitor CSN5i-3 in the presence and absence of Nedd8-conjugated cullin-RING ubiquitin ligase (N8~CRL) substrate or Nedd8 alone as well as structures of a CSN with inactive, doubly mutated CSN5 (CSN5DM) in complex with N8~CRL. The key advances are two-fold: (1) the use of CSN5DM enabled determination, for the first time, of a high-resolution structure of CSN in a stable 'pre-catalytic' complex with N8~CRL substrate. This allowed the authors to determine the molecular details of enzyme-substrate interaction and how it relates to catalytic mechanism. While this is an important advance, it is more in the category of incremental as opposed to transformative in terms of its likely influence on future work on CSN. (2) elucidation of the detailed mechanism-of-action of CSN5i-3. The authors make the very interesting observation that the inhibitor, which binds the catalytic zinc in the active site, also makes contact with the N8~CRL substrate. As a consequence the inhibitor binds to CSN much more tightly in the presence of substrate or even Nedd8 by itself, and this is shown directly with biophysical experiments. Figure 3 is the heart of the paper. The experiments therein provide an elegant explanation for the uncompetitive inhibition previously

reported for CSN5i-3. More importantly, the authors extrapolate from this finding to propose what appears to be a novel concept: orthostatic molecular glue (OMG) inhibitors. They point out that OMG inhibitors could potentially be used in circumstances where substrate binds at both an active site and exosite, and could be used to generate substrate-selective inhibitors. It is an intriguing idea and potentially important from a conceptual standpoint. It is this set of observations that justify publication in Nature in the opinion of this reviewer

Detailed comments:

1. Line 138: Fig 2b doesn't explicitly show the interactions that are discussed in the text.
2. Line 203-204: please state what the distance is. It is not given in primary text or figure legend. A reader shouldn't have to dig too deeply to get this information, which is important for assessing the claim made by the authors.
3. Line 255: this is worded poorly. To my knowledge reference 24 didn't report CSN5i-3 to act as a covalent inhibitor – it was shown to mimic the property of a covalent inhibitor in that it is slowly reversible.
4. Line 288: The effect of CSN5i-3 on CRL substrate receptor stability/levels is puzzling. Is it possible that the effect is due to failure to inactivate N8~CRL resulting in SR autoubiquitination? Are the surfaces that would recruit E2~Ub even accessible in the CSN-N8~CRL complex?
5. Line 289: on UniProt FBL12 is presented as an alternative name for the product of the FBXL12 gene, not FXL12
6. Lines 372-4: it would be preferable to see a figure that explicitly shows the interface with the RBX1 residues that were mutated called out, so it is possible to see how they are interacting with Cul1-WHB
7. Fig 3f: is it really the case that the $\Delta 71-76$ not cause a detectable migration shift on SDS-PAGE?
8. Fig 4a: This figure is very confusing. Why are there two rings of SRs and why do 2 and 6 point to the same dotted circle? Is this taken to mean that there is 100% overlap in the SRs pulled down by CSN2 and CS6? i.e. there was no SR pulled down by only one of them? While the colored circles may be graphically pretty, they are not particularly useful. Simply having a number would be more useful. E.g., is it correct to interpret that CSN2 and/or CSN6 pull down 23 DCAF proteins? It would be nice to know that without having to manually count brown circles.
9. Fig. 4b: It would be useful if text of SRs is color-coded according to what Cullin they bind (or is shown in a tinted box). This would make it more straightforward to rapidly assess, visually, whether there is a bias of effect towards one cullin or another. Also the figure needs a color scale.
10. There are examples of inhibitors that come close to the orthostatic molecular glue MoA as defined by the authors but don't satisfy every criterion. Two that come to mind are the MTA-dependent PRMT5 inhibitors and KIF18A inhibitors (e.g. sovilnesib). Both have "glue-like" properties in terms of forming contacts with the target and another molecule to convert a normally dynamic interface into a highly stable one. The authors might want to make some brief comments about this.

Referee #3

(Remarks to the Author)

In this manuscript "An Orthosteric Molecular Glue Inhibitor Locks COP9 Signalosome in Action" by Shi et al., the authors elegantly uncover the molecular mechanism of CSN5i-3 through biochemical, biophysical, and structural studies revealing it both inhibits the activity of the COP9 signalosome and strengthens the interaction with NEDD8. The authors additionally show for the first time nicely resolved precatalytic structures of CSN with N8-CRL1, CRL2, CRL3, CRL4A, and CRL5. Cryo-EM structures of all these complexes greatly advance the field's understanding of CSN's catalytic function and how it both differs and yet is similar in interacting with various CRL family members. The data represented here are robust, of the highest quality and caliber, and demonstrate a heroic structural feat to have determined each complex in physiologically relevant states of activity. I commend the authors on the impressive and rigorous work.

I have two high-level comments for the authors:

1. The manuscript reads as two separate stories - the first a story about the molecular glue inhibitor and second the story about CSN-N8-CRLs structures and activity. In reading the first part, I wanted to see the structures in Figure 5 before seeing the structure with the molecular glue inhibitor. Since this is a molecular glue story of augmentation of a native interaction, it is important to set the stage with all the information about the native interaction first then demonstrate how it is changed.
2. It is difficult to understand how much glue-ing there is with this compound in the context of the full CSN + N8-CRL complex or if the enhanced biochemical activity of the inhibitor is due to better trapping of the compound, leading to it having a more prolonged off-rate in the presence of the N8-CRL complex.

Specific comments:

1. It is striking in Figure 3a and b that CSN5i-3 can enhance the interaction of CSN5-6 and CSN with NEDD8 from undetectable to Kds of 930nM and 130nM. These however are still a ~log-fold order off the observed activity shown in Figure 1h (IC50 29nM). A significant discrepancy is that the activity assay is done with N8-CRL1 + CSN, not NEDD8 alone which was used in the enhanced binding study. Extended Data Figure 3 shows that the larger complex (N8-CRL1 + CSN DM) has a Kd of 39nM, which now is similar to the IC50. Does this indicate the compound is not significantly enhancing the larger complex interaction with CSN, only the NEDD8? The missing experiment here if possible to do would be BLI with ligand: N8-CRL1, Analyte: CSN, + CSN5i-3. If there is not a significant shift in affinity, then it suggests more of a mechanism of compound trapping and alteration of koff.
2. Figure 4 - are losses in proteins due to changes in RNA levels and/or dependent upon proteasomal activity? It is quite confusing that some of the substrate receptors are decreased, and others are increased. I recommend moving these data to Extended figures and also include further controls such as RNA seq and/or compound +/- proteasomal inhibitors to understand what is driving the changes in protein level.

3. Extended Data Figure 4 - Suggest moving to be a main figure and then the schematic F being moved to the end of Figure 3.

4. There are other compounds that come to mind which are similar in concept to CSN5i-3 in that they enhance the Kd of a native interaction by acting as a molecular glue and also inhibit the activity of the target they are bound.

1. BRD4 monovalent or bivalent glue inhibitors which significantly enhance a weak native interaction with DCAF16. The glueing ultimately leads to BRD4 degradation.

<https://doi.org/10.1021/acschembio.2c00747>

<https://doi.org/10.1038/s41586-024-07089-6>

2. Molecular glue inhibitors between MEK:Raf and MEK:KSR complexes, including one from NestedTx in clinic:

<https://doi.org/10.1038/s41586-020-2760-4>

<https://doi.org/10.1158/2159-8290.CD-24-0139>

In addition, there is a review which has been published during the time this article was submitted summarizing therapeutics which will change the off-rates of natively interacting proteins, entitled Load and lock: An emerging class of therapeutics that influence macromolecular dissociation

<https://www.science.org/doi/10.1126/science.adx3595>

From the data presented in this manuscript, CSN5i-3 would seem to fall into this emerging class

Given all the above I am not certain it is an entirely new class of molecular glues and warrants a new category classification of glues (though I found OMG to be clever).

Small point:

1. The title suggests COP9 Signalosome is locked into action, which it is not.

Dr. Danette Daniels

Version 1:

Reviewer comments:

Referee #1

(Remarks to the Author)

I found the revised version excellent. This is a significant, insightful work. I have no further comments.

Referee #2

(Remarks to the Author)

I believe that the authors have adequately addressed my comments from the first cycle of review, which were mainly aimed towards enhancing the clarity of the overall presentation and addressing relevant points of discussion. I have to say I love the idea of making an orthostatic inhibitor of **TEXT REDACTED** as suggested in the author's response. I'm left to wonder if it was AndyChan, Hans Clevers, or someone else? I don't think it was me... (Ray Deshaies)

Referee #3

(Remarks to the Author)

Kudos to all the authors for the significant revision to this manuscript! Clearly there has been tremendous effort and thoughtfulness put into rewriting, reorganization, and additional experiments to fully understand the mechanism of action. These changes have resulted in a unified story, improved clarity, and demonstration of distinct mechanism of this new class of molecular glues (OMGs). I have no further recommended changes and look forward to seeing this being finalized for publication.

Congratulations to all involved on this remarkable work and these discoveries -

Dr. Danette L. Daniels

POINT-TO-POINT RESPONSES TO REVIEWERS' COMMENTS

Introduction

We would like to first express our gratitude to all three reviewers for their thoughtful reviews and constructive feedback. We have followed the suggestions made by the reviewers and revised our manuscripts with some major changes summarized below and other edits detailed in the rest of this Point-to-Point Responses document.

1. Following Reviewer #3's suggestion, we have restructured the paper to present the pre-catalytic state structure of CSN^{DM}-N8~CRL1 before describing our findings on the unique mechanism of action of CSN5i-3 as an orthosteric molecular glue (OMG) inhibitor. This revision required reorganizing most figures and rewriting transitions between sections. We are very pleased with the outcome, as the revised narrative flow is more natural and coherent. This restructuring also effectively addresses Reviewer #2's concern about two poorly connected themes in the manuscript with disparate impact.
2. We have included new data to validate the molecular glue effect of CSN5i-3 on the interaction between CSN and the full N8~CRL1 substrate. This addresses one of the major questions raised by Reviewer #3. We are grateful to the reviewer for prompting these experiments, which significantly strengthen our conclusions.
3. Following editorial advice and in response to the molecule glues examples highlighted by the reviewers, we have revised the title, abstract and discussion section of our manuscript to better emphasize the unique properties of OMG inhibitors. We believe that these changes help clarify the conceptual distinction between OMG inhibitors and other types of small molecule enzyme inhibitors with molecular glue properties.

We have also addressed each specific point raised by the reviewers, both major and minor, as detailed below.

Referees' comments:

Referee #1 (Remarks to the Author):

Here the authors essentially discover an allosteric mechanism through which orthosteric inhibitors not only block substrate binding at the active site but enhance their potency by allosterically altering the exosite conformation, leading to exposing the surface of the exosite in a binding-favored state, to which the substrate binds. In turn, substrate binding at the exosite cooperatively enhances their inhibitory action. They beautifully show that CSN5i-3, an orthosteric inhibitor of the COP9 signalosome (CSN), which catalyzes NEDD8 deconjugation from the cullin-RING ubiquitin ligases (CRLs), thus promotes the substrate-enzyme assembly, which cooperatively stabilizes its own interaction at the CSN's active site. Whereas the authors highlight it as unlikely, I would not call their observation as such. Allosteric effects are bi-directional. They are cooperative. An allosteric binding partner can modulate – enhance or diminish– the efficacy of the orthosteric ligand (inhibitor), while concomitantly, the orthosteric

ligand (inhibitor) can modulate the efficacy of the allosteric ligand.

I find this work exciting not only because of the novelty and its potential impact on the crucial CSN system, but also because of its potential broad implications. Any binding at any site on the surface will stimulate a shift of the protein ensembles. Here the shift elicited by CSN5i-3 leads to conformational changes at the NEDD8-binding exosite, hindering CSN5 cleavage of NEDD8 modification from Cullin-RING E3 ubiquitin ligases (CRLs), thus trapping them in their neddylated, active state. These conformational changes are crucial, as they expose the occluded loop in the CSN holoenzyme structure. At the same time, the shift of the ensemble generated by the allosteric binding of the substrate cooperatively increases the efficacy of CSN5i-3. Thus, they discovered that CSN5i-3 functions as both an orthosteric inhibitor of the iso-peptidase target, and a stabilizer of the enzyme-substrate complex, which led them to coin orthosteric molecular glue (OMG) inhibitors. I would add that its efficacy is further strengthened allosterically.

I have only one comment: CSN does not undergo major conformational changes upon substrate engagement as the paper notes. Instead, the conformational changes were likely promoted by the inhibitor binding and are captured by the substrate. They may or may not be observed in the pre-substrate binding state, depending on the relative stability of the state – stable or sparse. A limited resolution may suggest a broad distribution, that is a sparse state.

We sincerely thank the reviewer for his/her positive evaluation of our work and, especially, his/her insightful comments on the “bidirectional” cooperative nature of compound-enhanced protein-protein interaction and the reciprocal impact on the efficacy of the (inhibitory) ligand. To strengthen our story, we have emphasized this aspect of our observations in our revision, which is also suggested by the editor. We also agree with the reviewer that the limited resolution of our structure might have prevented us from detecting a sparse state of the enzyme, which does undergo subtle but critical conformational changes upon inhibitor binding. Such an allosteric mechanism could work in synergy with the molecular glue mechanism to stabilize the ternary enzyme-substrate-inhibitor assembly.

Referee #2 (Remarks to the Author):

The manuscript by Shi and colleagues reports structures of CSN in complex with its inhibitor CSN5i-3 in the presence and absence of Nedd8-conjugated cullin-RING ubiquitin ligase (N8~CRL) substrate or Nedd8 alone as well as structures of a CSN with inactive, doubly mutated CSN5 (CSN5DM) in complex with N8~CRL. The key advances are two-fold: (1) the use of CSN5DM enabled determination, for the first time, of a high-resolution structure of CSN in a stable ‘pre-catalytic’ complex with N8~CRL substrate. This allowed the authors to determine the molecular details of enzyme-substrate interaction and how it relates to catalytic mechanism. While this is an important advance, it is more in the category of incremental as opposed to transformative in terms of its likely influence on future work on CSN. (2) elucidation of the detailed mechanism-of-action of CSN5i-3. The authors make the very interesting observation that the inhibitor, which binds the catalytic zinc in the active site, also makes contact with the N8~CRL substrate. As a consequence the inhibitor binds to CSN much more tightly in the presence of substrate or even Nedd8 by itself, and this is shown directly with biophysical

experiments. Figure 3 is the heart of the paper. The experiments therein provide an elegant explanation for the uncompetitive inhibition previously reported for CSN5i-3. More importantly, the authors extrapolate from this finding to propose what appears to be a novel concept: orthostatic molecular glue (OMG) inhibitors. They point out that OMG inhibitors could potentially be used in circumstances where substrate binds at both an active site and exosite, and could be used to generate substrate-selective inhibitors. It is an intriguing idea and potentially important from a conceptual standpoint. It is this set of observations that **justify publication in Nature** in the opinion of this reviewer.

We truly appreciate the concise summary and fair evaluation of our study by the reviewer, who is spot-on by pointing out that “Figure 3 is the heart of the paper”. We also consider the concept of the OMG inhibitors has a broader and more far-reaching impact than the CSN-specific enzyme-substrate interaction findings. In fact, the excitement around the OMG inhibitor concept was echoed by colleagues from both academia and industry, whom we have recently shared our results with. As pointed out by a retiring CSO from a large pharmaceutical company, the OMG inhibitor concept can be readily applied to the **TEXT REDACTED** orthosteric inhibitors, which have frustrated almost every major pharmaceutical companies worldwide in the past two decades due to their substrate-indiscriminating effects. Although these inhibitors can effectively **TEXT REDACTED**, they showed various side effects because they also block the processing of other physiological substrate proteins **TEXT REDACTED**. With substrate-dependent potency, a second generation **TEXT REDACTED** OMG inhibitors could be developed to gain efficacy towards only **TEXT REDACTED**

Encouraged by the comments from this reviewer as well as the suggestion made by Reviewer 3, we have now re-organized the manuscript to describe the pre-catalytic state structure first before the mechanistic analyses of the inhibitor. This new arrangement streamlines the studies and leverages the enzyme-substrate interaction results to substantiate the orthosteric nature of the inhibitor. It also makes the theme of the story more focused and coherent. We, again, thank the reviewer for his/her constructive remarks.

Detailed comments:

1. Line 138: Fig 2b doesn't explicitly show the interactions that are discussed in the text.

In the revised manuscript, the original Fig 2b has been moved to Fig 4b. We have also revised the sentence and refer to Fig 4b-d to be more accurate.

2. Line 203-204: please state what the distance is. It is not given in primary text or figure legend. A reader shouldn't have to dig too deeply to get this information, which is important for assessing the claim made by the authors.

We thank the reviewer for pointing out this oversight. We have now included the distance threshold for DSSO cross-linking (“~30 Å) in the primary text: “In both cases, the opposite effects can be explained by the maximal inter-lysine distance allowed for efficient XL (30 Å) (Fig. 4g).” In addition to the required residue-to-residue proximity, the formation of a cross-link

depends on several other factors including their solvent accessibility and chemical microenvironments. Moreover, their identification relies on the abundance and MS fragmentation of the resulting cross-linked peptides. The relevant figure is now Fig 4g.

3. Line 255: this is worded poorly. To my knowledge reference 24 didn't report CSN5i-3 to act as a covalent inhibitor – it was shown to mimic the property of a covalent inhibitor in that it is slowly reversible.

The sentence is indeed worded improperly. We have now changed the sentence, which reads “As an uncompetitive inhibitor, CSN5i-3 shows a nearly irreversible effect in the cell, analogous to a covalent inhibitor”.

4. Line 288: The effect of CSN5i-3 on CRL substrate receptor stability/levels is puzzling. Is it possible that the effect is due to failure to inactivate N8~CRL resulting in SR autoubiquitination? Are the surfaces that would recruit E2~Ub even accessible in the CSN-N8~CRL complex?

The reviewer raises a very good question. First of all, our finding that CSN5i-3 destabilizes some CRL SRs is consistent with previously published results (*e.g.*, reference #24). However, as pointed out by the reviewer, the surface of RBX1 that is involved in recruiting ubiquitin-charged E2 should be blocked by CSN in the CSN-N8-CRL-SR complexes. If the destabilizing effect is indeed due to failure to inactivate N8~CRL, which results in SR autoubiquitination, it must happen after CSN is dislodged. It is plausible that the engagement of certain SRs, either alone or together with their physiological substrates, with the CRL catalytic scaffold (cullin-RING) might be incompatible with the CSN5i-3 stabilized CSN-N8~CRL complex due to their specific shape, size, or spatial arrangement. Therefore, CSN5i-3 might destabilize the formation of these CSN-N8~CRL-SR assemblies. Of note, most studies aimed at dissecting the interplay among CSN, N8~CRL, SRs, substrates, and CAND1 (the proposed SR exchange factor) have been carried out *in vitro*. In our humble opinion, it remains unclear when and how these components interact with each other in the cell and in response to cellular signals triggering substrate ubiquitination and degradation. The answer to the burning question raised by the reviewer requires future studies, which, hopefully, can be designed based on our results.

5. Line 289: on UniProt FBL12 is presented as an alternative name for the product of the FBXL12 gene, not FXL12.

For Fig 4b, protein names have now been replaced with the formal gene name. However, CSN subunit names have been retained as CSN1-8 (instead of “COPS1-8”) for consistency in the text.

6. Lines 372-4: it would be preferable to see a figure that explicitly shows the interface with the RBX1 residues that were mutated called out, so it is possible to see how they are interacting with Cul1-WHB

The reviewer made a good point. In the original manuscript, the three aromatic residues at the interface were described in an earlier section, which made it inconvenient to find for the readers. Now that we have rearranged the sections, the CUL1-WHB-RBX1 interface is described in the same section and consecutive figure panel as the mutational results (“Pre-catalytic state protein-

protein interaction” and Fig 2d, 2e). We have also called out the specific mutations in the associated figure legend.

7. Fig 3f: is it really the case that the $\Delta 71-76$ not cause a detectable migration shift on SDS-PAGE?

In the revised manuscript, Fig 3f becomes Fig 5f. Although a six amino acids deletion is expected to affect the migration of NEDD8 (8 kD) on a SDS-PAGE gel, there are two factors at play that diminish the expected effect. First, in this assay, NEDD8 is fused to Venus (a yellow-shifted GFP mutant with a molecular weight of 27 kD). The increased molecular weight decreased the migration difference between the WT and the truncation mutant. Second, in order to visualize the protein bands via the fluorescent signal of Venus, we did not boil and denature the protein samples before loading. This affected the migration position of the fusion protein. The WT and mutant fusion proteins all migrated above the 48 kD protein marker.

To further confirm that the truncation mutant is indeed missing amino acid 71 to 76 in NEDD8, we have sequenced the construct again. We have also run a separate SDS-PAGE gel with the WT and mutant fusion proteins by boiling the samples and visualizing the protein bands by Coomassie staining. Despite being subtle, the protein bands do show a slightly different migration pattern (see figure below). They also migrate similarly as the 35 kD marker. We greatly appreciate the reviewer’s careful attention to detail and sharp observation.

8. Fig 4a: This figure is very confusing. Why are there two rings of SRs and why do 2 and 6 point to the same dotted circle? Is this taken to mean that there is 100% overlap in the SRs pulled down by CSN2 and CSN6? i.e. there was no SR pulled down by only one of them? While the colored circles may be graphically pretty, they are not particularly useful. Simply having a number would be more useful. E.g., is it correct to interpret that CSN2 and/or CSN6 pull down 23 DCAF proteins? It would be nice to know that without having to manually count brown circles.

We thank the reviewer for pointing this out. For clarity, we have added the number of SRs corresponding to each CRL subfamily. The detailed identification and quantification of SRs shown in the figure for CSN2 and CSN6 is reported in Extended Data Table 3b. As illustrated, CSN2 and CSN6 have co-purified the same SR proteins which were identified in at least one biological replicate. Given the significant overlap of SRs co-purified with CSN2 and CSN6

(153/159) in at least two replicates, we have opted to show the total coverage of SRs across both affinity purifications.

9. Fig. 4b: It would be useful if text of SRs is color-coded according to what Cullin they bind (or is shown in a tinted box). This would make it more straightforward to rapidly assess, visually, whether there is a bias of effect towards one cullin or another. Also the figure needs a color scale.

We thank the reviewer for the suggestions. The SRs in Fig 6b (which was originally Fig 4b) have been designated with colored circles corresponding to their CRL subfamily as shown in Fig 6a (which was the original Fig 4a). Additionally, a color scale has been provided.

10. There are examples of inhibitors that come close to the orthostatic molecular glue MoA as defined by the authors but don't satisfy every criterion. Two that come to mind are the MTA-dependent PRMT5 inhibitors and KIF18A inhibitors (e.g. sovilnesib). Both have "glue-like" properties in terms of forming contacts with the target and another molecule to convert a normally dynamic interface into a highly stable one. The authors might want to make some brief comments about this.

We thank the reviewer and Reviewer #3 for highlighting these and other interesting examples of molecular glue compounds. Based on the feedback from both reviewers and guidance from the editor, we recognize the need to better emphasize the thesis of this work, which is focused on the classification of OMG inhibitors as a novel class of substrate-dependent orthosteric inhibitors. The translational value of this concept lies in enabling the development of orthosteric inhibitors of enzymes with substrate selectivity. The MTA-dependent PRMT5 inhibitors elegantly exploit a metabolic vulnerability to achieve context-dependent inhibition. However, MTA is not a substrate of PRMT5. The KIF18A inhibitors Sovilnesib and its precursor BTB-1 bind to an allosteric pocket of the kinesin-like protein and has been proposed to act as molecular glues that immobilizes the motor protein on microtubules. While these examples share the common theme of the molecular glue mechanisms, they do not necessarily align with the specific concept of OMG inhibitors. Please see our response to Reviewer #3 point 4 for further discussion.

To clarify these distinctions, we have revised the title, abstract, and discussion section of the manuscript and cited the relevant publications to emphasize the conceptual differences of OMG inhibitors.

Referee #3 (Remarks to the Author):

In this manuscript "An Orthosteric Molecular Glue Inhibitor Locks COP9 Signalosome in Action" by Shi et al., the authors elegantly uncover the molecular mechanism of CSN5i-3 through biochemical, biophysical, and structural studies revealing it both inhibits the activity of the COP9 signalosome and strengthens the interaction with NEDD8. The authors additionally show for the first time nicely resolved precatalytic structures of CSN with N8-CRL1, CRL2, CRL3, CRL4A, and CRL5. Cryo-EM structures of all these complexes greatly advance the

field's understanding of CSN's catalytic function and how it both differs and yet is similar in interacting with various CRL family members. The data represented here are robust, of the highest quality and caliber, and demonstrate a heroic structural feat to have determined each complex in physiologically relevant states of activity. I commend the authors on the impressive and rigorous work.

We are deeply grateful to the reviewer for her thorough evaluation and for providing such positive and constructive feedback.

I have two high-level comments for the authors:

1. The manuscript reads as two separate stories - the first a story about the molecular glue inhibitor and second the story about CSN-N8-CRLs structures and activity. In reading the first part, I wanted to see the structures in Figure 5 before seeing the structure with the molecular glue inhibitor. Since this is a molecular glue story of augmentation of a native interaction, it is important to set the stage with all the information about the native interaction first then demonstrate how it is changed.

We wholeheartedly agree with the reviewer on this brilliant suggestion! By presenting the native interaction first, the new arrangement indeed streamlines the story with a coherent theme and sets the stage for the OMG inhibitor finding, which is the central message of the manuscript. It makes the development of the story more natural and highlights better the more important part of the story. Following this suggestion made by the reviewer, we have now re-organized the manuscript by describing the pre-catalytic state of the complex first before introducing our findings of the inhibitor. We have also re-written the abstract and changed the title to be consistent with the story flow and the central theme of the main text. Once again, we truly appreciate the reviewer's comment, which is exceedingly constructive.

2. It is difficult to understand how much glue-ing there is with this compound in the context of the full CSN + N8-CRL complex or if the enhanced biochemical activity of the inhibitor is due to better trapping of the compound, leading to it having a more prolonged off-rate in the presence of the N8-CRL complex.

This is also a very insightful question from the reviewer regarding the unique mechanism of action of the OMG inhibitor. On the one hand, the compound might show a better efficacy by stabilizing the non-productive substrate-enzyme interaction, thereby, preventing the engagement of another substrate molecule with the enzyme. On the other hand, the non-productive substrate-enzyme assembly might increase the potency of the inhibitory compound by trapping it at the active site and impeding its release. Given the "bi-directional" and cooperative nature of the ternary complex formation, as commented by Reviewer #1, we believe that both mechanisms contribute to the improved activity of the inhibitory compound. While the substrate-dependent high potency of the compound, by and large, reflects its higher affinity to the enzyme-substrate complex than to the free enzyme (most likely a koff effect), we have now included new data to support the notion that the OMG inhibitor indeed also stabilizes the non-productive substrate-enzyme assembly in the context of the full CSN + N8~CRL complex (see our response to Specific Comment 1 below).

Specific comments:

1. It is striking in Figure 3a and b that CSN5i-3 can enhance the interaction of CSN5-6 and CSN with NEDD8 from undetectable to Kds of 930nM and 130nM. These however are still a ~log-fold order off the observed activity shown in Figure 1h (IC50 29nM). A significant discrepancy is that the activity assay is done with N8-CRL1 + CSN, not NEDD8 alone which was used in the enhanced binding study. Extended Data Figure 3 shows that the larger complex (N8-CRL1 + CSN DM) has a Kd of 39nM, which now is similar to the IC50. Does this indicate the compound is not significantly enhancing the larger complex interaction with CSN, only the NEDD8? The missing experiment here if possible to do would be BLI with ligand: N8-CRL1, Analyte: CSN, + CSN5i-3. If there is not a significant shift in affinity, then it suggests more of a mechanism of compound trapping and alteration of koff.

We thank the reviewer for raising this interesting issue. To quantify the molecular glue effect of the inhibitor on CSN-N8~CRL1 interaction, ideally one would need to measure the dissociation constant of CSN and N8~CRL1 in the absence and presence of the inhibitor. This experiment, however, is tricky because CSN would hydrolyze N8~CRL1 in the absence of the compound. To bypass this problem, one could replace CSN with the catalytically dead mutant (CSN^{DM}). However, there are obvious caveats associated with this strategy as well. First of all, the mutant bears two different amino acids at the active site in comparison to the wild type. Second, it lacks the catalytic zinc ion. More importantly, the affinity comparison for detecting the molecular glue effect should, strictly speaking, be made between (1) the Kd value of CSN binding to N8~CRL1 in the presence of the compound and (2) the Kd value of CSN binding to N8~CRL1 with the isopeptide linkage removed from the active site. Such a comparison is analogous to other molecular glue systems, in which two proteins interact with each other through an imperfect interface, which is spatially complemented by the small molecule.

As an alternative to the aforementioned experiments, we did manage to find a solution to address the question raised by the reviewer. As we have described in our original manuscript, the highly potent CSN5i-3 compound was derived from an HTS hit compound, CSN5i-1a. Both compounds bind free CSN5 with a similar single digit μ M affinity. Yet only CSN5i-3 shows substrate-dependent potency in the low nM range (reference #23, new Fig 3d, 3e, ED Fig 3b, 3c, and Fig 5i, 5j). Our structural results reveal that the unique distal difluoromethyl pyrazole moiety possessed by CSN5i-3 makes direct contact with NEDD8 and is responsible for the molecular glue effect. At a saturating concentration, both compounds would occlude the isopeptide linkage from the CSN5 active site. If CSN5i-3 indeed stabilizes substrate-enzyme interaction in the context of the full CSN-N8~CRL1 complex, one would expect that the Kd values of CSN binding to N8~CRL1 in the presence of the two individual compounds would be significantly different. As shown in the new ED Fig 4d and 4e, this is indeed the case. While the affinity of N8~CRL1 towards CSN5i-1a-bound CSN is in the single digit μ M range (1.8 μ M), the corresponding number for CSN5i-3 is 26 nM. This difference almost perfectly matches the difference between CSN5i-3's affinity to free CSN (4.7 μ M) and its potency in inhibiting the enzyme (29 nM).

We apologize for this rather lengthy response but sincerely thank the reviewer for prompting us to perform this experiment, the result of which has truly strengthened the story.

2. Figure 4 - are losses in proteins due to changes in RNA levels and/or dependent upon proteasomal activity? It is quite confusing that some of the substrate receptors are decreased, and others are increased. I recommend moving these data to Extended figures and also include further controls such as RNA seq and/or compound +/- proteasomal inhibitors to understand what is driving the changes in protein level.

Following the reviewer's suggestion, we have included new data to show that MG132, a proteasomal inhibitor, can block the effects of CSN5i-3 in destabilizing the two representative substrate receptors. We have moved the original Fig 4c and 4d, together with these new control results, to Extended Data figure (new ED Fig 4f-h). Nevertheless, we decide to retain the overall proteomics data in the new Fig 6 for two reasons. First, the compound-enhanced binding of CSN to a fraction of the CRL-SRs is consistent with the molecular glue effect of CSN5i-3, adding supporting evidence from cell-based analysis. Second, this unexpected result is informative to the field, where it is widely believed that the inhibitor would destabilize the majority, if not all, SRs. Presenting the proteomics data in the final main figure would help highlight an open question to be addressed by future studies.

3. Extended Data Figure 4 - Suggest moving to be a main figure and then the schematic F being moved to the end of Figure 3.

We appreciate the suggestions made by the reviewer. Due to space limitation, we find it difficult to include the original ED Fig 4a-d to a new main figure. However, we do agree with the reviewer that including the schematic ED Fig 4f panel in a main figure (new Fig 6) will help strengthen the take-home message of the entire story.

4. There are other compounds that come to mind which are similar in concept to CSN5i-3 in that they enhance the K_d of a native interaction by acting as a molecular glue and also inhibit the activity of the target they are bound.

1. BRD4 monovalent or bivalent glue inhibitors which significantly enhance a weak native interaction with DCAF16. The glueing ultimately leads to BRD4 degradation.

<https://doi.org/10.1021/acscchembio.2c00747>

<https://doi.org/10.1038/s41586-024-07089-6>

2. Molecular glue inhibitors between MEK:Raf and MEK:KSR complexes, including one from NestedTx in clinic:

<https://doi.org/10.1038/s41586-020-2760-4>

<https://doi.org/10.1158/2159-8290.CD-24-0139>

In addition, there is a review which has been published during the time this article was submitted summarizing therapeutics which will change the off-rates of natively interacting proteins, entitled Load and lock: An emerging class of therapeutics that influence macromolecular dissociation

<https://www.science.org/doi/10.1126/science.adx3595>

From the data presented in this manuscript, CSN5i-3 would seem to fall into this emerging class

Given all the above I am not certain it is an entirely new class of molecular glues and warrants a

new category classification of glues (though I found OMG to be clever).

We thank the reviewer and Reviewer #2 for highlighting these and other interesting examples of molecular glue compounds. Based on these comments made by both reviewers and guidance from the editor, we recognize the need to better emphasize the central thesis of this work, which is focused on the classification of OMG inhibitors as a novel class of substrate-dependent orthosteric inhibitors, instead of molecular glues. We would like to underline two key aspects of this concept, which the original manuscript might have failed to clearly convey.

First, we share the excitement on molecule glues with the authors of the review article entitled “Load and lock, an emerging class of therapeutics that influence macromolecular dissociation”. However, prospective discovery of novel molecular glue compounds has been highly challenging. For example, in the field of targeted protein degradation, there is currently no well-established and robust strategy to select a ubiquitin ligase out of the 600 human E3s to pair with a target of interest as a neo-substrate before a compound screen campaign can be started. By contrast, rational development of conventional orthosteric inhibitors is relatively “straightforward”, benefiting from a well-defined pocket on a single target. It provides a more tractable starting point for developing an OMG inhibitor. By introducing a protein-protein interaction assay in parallel to an enzyme inhibition assay, one would have the opportunity to convert a conventional orthosteric inhibitor into an OMG inhibitor.

Second, the most distinctive property of OMG inhibitors, which set them apart from conventional orthosteric inhibitors, is their modest affinity towards the free enzyme target and their high potency in a substrate-dependent manner. This property opens new opportunities for re-engineering existing orthosteric inhibitors, such as **TEXT REDACTED** (see our response to Reviewer #2’s general comment), which have failed in clinical trials due to their broad impact to all substrates of the enzyme target, to achieve substrate selectivity. Notably, this characteristic of OMG inhibitors arises from the cooperativity inherent in molecular glue systems, where a compound need not bind with high affinity to either protein partner (reference #30).

Together, these two characteristics distinguish OMG inhibitors from other classes of inhibitors and molecule glue compounds highlighted by the two reviewers. While the molecular glue mechanism contributes to the mode of action of OMG inhibitors, it does not encompass their entire conceptual framework.

Specifically, in the case of the BRD4 monovalent or bivalent glue degraders, the compounds were not rationally developed as DCAF16-dependent enzyme inhibitors, and prospective discovery of degraders acting through similar mechanisms remains highly challenging. The molecular glue compounds trametinib and NST-628 inhibit MEK by stabilizing the MEK-KSR and MEK-RAF complexes, respectively. They are not orthosteric MEK inhibitors but instead bind to an allosteric site distinct from the ATP binding pocket. Moreover, to our knowledge, no direct comparison has been made between the potencies of these compound and their affinities to free MEK.

To clarify these distinctions, we have revised the title, abstract, and discussion section of the manuscript and cited the relevant publications to emphasize the conceptual differences of OMG

inhibitors. While we are delighted that the OMG nomenclature resonated with the reviewer, we hope the discussion above has made a compelling case for why this new concept merits its own classification.

Small point:

1. The title suggests COP9 Signalosome is locked into action, which it is not.

This is a valid point! In light of other suggestions made by the reviewer, we have changed the title of the manuscript to better highlight the central theme of the story. We thank the reviewer for raising this “small” but important point.

POINT-TO-POINT RESPONSES TO REVIEWERS' COMMENTS

Referee #1 (Remarks to the Author):

I found the revised version excellent. This is a significant, insightful work. I have no further comments.

We once again thank Referee #1 for his/her time and effort spent in reviewing our manuscript.

Referee #2 (Remarks to the Author):

I believe that the authors have adequately addressed my comments from the first cycle of review, which were mainly aimed towards enhancing the clarity of the overall presentation and addressing relevant points of discussion. I have to say I love the idea of making an orthostatic inhibitor of **TEXT REDACTED** as suggested in the author's response. I'm left to wonder if it was Andy

Chan, Hans Clevers, or someone else? I don't think it was me... (Ray Deshaies)

We are grateful to Referee #2, Dr. Deshaies, for his comments and suggestions made in the first round of review. The idea of applying the OMG inhibitor concept to **TEXT REDACTED**. We are excited about this opportunity and are actively working on the project.

Referee #3 (Remarks to the Author):

Kudos to all the authors for the significant revision to this manuscript! Clearly there has been tremendous effort and thoughtfulness put into rewriting, reorganization, and additional experiments to fully understand the mechanism of action. These changes have resulted in a unified story, improved clarity, and demonstration of distinct mechanism of this new class of molecular glues (OMGs). I have no further recommended changes and look forward to seeing this being finalized for publication.

Congratulations to all involved on this remarkable work and these discoveries -
Dr. Danette L. Daniels

We sincerely appreciate Dr. Daniels' exceptionally constructive suggestions made in the last round of peer review. We wholeheartedly agree that the story has been significantly improved with the suggested changes implemented. We are looking forward to the feedback from other colleagues in the field.